# Parameter-Free Hypergraph Neural Network for Few-Shot Node Classification

**Chaewoon Bae**    **Doyun Choi**    **Jaehyun Lee**    **Jaemin Yoo**
School of Electrical Engineering
Korea Advanced Institute of Science and Technology (KAIST)
{chaewoon.bae, doyun.choi, jaehyun.lee, jaemin}@kaist.ac.kr

## Abstract

Few-shot node classification on hypergraphs requires models that generalize from scarce labels while capturing high-order structures. Existing hypergraph neural networks (HNNs) effectively encode such structures but often suffer from overfitting and scalability issues due to complex, black-box architectures. In this work, we propose ZEN (Zero-Parameter Hypergraph Neural Network), a fully linear and parameter-free model that achieves both expressiveness and efficiency. Built upon a unified formulation of linearized HNNs, ZEN introduces a tractable closed-form solution for the weight matrix and a redundancy-aware propagation scheme to avoid iterative training and to eliminate redundant self-information. On 11 real-world hypergraph benchmarks, ZEN consistently outperforms eight baseline models in classification accuracy while achieving up to $696\times$ speedups over the fastest competitor. Moreover, the decision process of ZEN is fully interpretable, providing insights into the characteristic of a dataset. Our code and datasets are fully available at https://github.com/chaewoonbae/ZEN.

## 1    Introduction

Many real-world datasets are naturally modeled as hypergraphs, which generalize ordinary graphs by capturing higher-order interactions involving more than two nodes [1, 13, 16]. Unlike traditional graphs that represent pairwise relationships, hypergraphs allow for the modeling of multilateral connections, making them particularly suitable for complex data structures. Hypergraphs are especially useful in representing multifaceted relationships across a range of domains, such as document-word associations, gene-disease correlations, and user-item interactions [6, 9, 29, 33].

Few-shot node classification—predicting node labels with only a handful of annotated examples—is a fundamental yet challenging task, especially in the context of hypergraphs [24, 34]. The complexity of higher-order structures, combined with the scarcity of labeled data, makes it difficult to design models that are both generalizable and efficient. Although a number of hypergraph neural networks (HNNs) have been proposed to effectively capture high-order relationships between nodes, many of them rely on complex, nonlinear architectures with numerous parameters. Such models often suffer from overfitting and poor scalability in few-shot scenarios [21, 32].

In contrast, linear models offer strong generalization, low complexity, and are particularly effective in few-shot scenarios [5, 18, 20, 25]. Despite these advantages, existing work on HNNs has largely overlooked fully linear formulations, likely due to the perception that linear models cannot sufficiently capture the structural richness of hypergraphs. This motivates us to explore if a carefully designed linear HNN can overcome these limitations while preserving both expressiveness and efficiency.

In this paper, we propose ZEN (Zero-Parameter Hypergraph Neural Network), a model designed to capture high-order relationships while maintaining strong generalization ability and scalability. We begin by linearizing existing HNN models, yielding a general form characterized by a single

propagation matrix and a single weight matrix. This formulation enables us to design two optimization strategies tailored to this structure: *tractable closed-form solution* (TCS) for the weight matrix and *redundancy-aware propagation* (RAP). TCS allows us to avoid burdensome iterative training by deriving an efficient closed-form solution, while RAP eliminates redundant self-information across multi-hop propagation. ZEN achieves the best average rank in classification performance across 11 real-world hypergraphs, while also exhibiting exceptional computational efficiency.

To summarize, our main contributions are as follows:

- We present a general form of linearizing representative hypergraph neural networks, and exhibit similarities and differences between different HNNs excluding their nonlinearity. To the best of our knowledge, this is the first comprehensive study on linear HNNs.
- We derive a tractable closed-form approximation for the weight matrix, the only parameter in a linear HNN. Our solution eliminates the need for computing matrix pseudoinverses, significantly improving computational efficiency while maintaining high classification accuracy.
- We introduce *redundancy-aware propagation* that effectively reduces the structural overlap across multi-hop neighborhoods when building the propagation matrix. This leads to more efficient and expressive information aggregation, especially uesful for complex real hypergraphs.
- We conduct comprehensive experiments on 11 real-world hypergraphs and demonstrate that our method outperforms existing HNNs in few-shot classification while showing exceptional scalability; our ZEN is up to $696\times$ faster than the fastest competitor. In addition, we conduct a case study that shows the interpretability of ZEN arising from its linear decision process.

## 2 Problem definition and related work

### 2.1 Problem definition

We consider a hypergraph $\mathcal{G} = (\mathcal{V}, \mathcal{E})$, where $\mathcal{V}$ is the set of nodes and $\mathcal{E}$ is the set of hyperedges. The incidence matrix $\mathbf{H} \in \mathbb{R}^{|\mathcal{V}| \times |\mathcal{E}|}$ encodes hypergraph connectivity, with $\mathbf{H}_{ij} = 1$ if node $i$ belongs to hyperedge $j$, and 0 otherwise. Each node is associated with a feature vector, forming a feature matrix $\mathbf{X} \in \mathbb{R}^{|\mathcal{V}| \times d}$ where $d$ is the feature dimension. The class label for each node is encoded in a one-hot matrix $\mathbf{Y} \in \{0, 1\}^{|\mathcal{V}| \times c}$, where $c$ is the number of classes.

For node classification, we are given a set $\mathcal{V}_{\text{trn}} \subset \mathcal{V}$ of training nodes and a set $\mathcal{V}_{\text{test}} \subset \mathcal{V}$ of test nodes such that $\mathcal{V}_{\text{trn}} \cap \mathcal{V}_{\text{test}} = \emptyset$. We define two diagonal indicator matrices $\mathbf{D}_{\text{trn}} \in \mathbb{R}^{|\mathcal{V}| \times |\mathcal{V}|}$ and $\mathbf{D}_{\text{test}} \in \mathbb{R}^{|\mathcal{V}| \times |\mathcal{V}|}$, where $(\mathbf{D}_{\text{trn}})_{ii} = 1$ iff $i \in \mathcal{V}_{\text{trn}}$ and $(\mathbf{D}_{\text{test}})_{ii} = 1$ iff $i \in \mathcal{V}_{\text{test}}$. In the $k$-shot setting, we assume that exactly $k$ labeled nodes are available per class, i.e., $\text{tr}(\mathbf{D}_{\text{trn}}) = kc$. The goal of $k$-shot node classification is to find a model $f$ that produces prediction $\hat{\mathbf{Y}} \in \mathbb{R}^{|\mathcal{V}| \times c}$ such that:

$$\mathbf{D}_{\text{test}}\hat{\mathbf{Y}} \approx \mathbf{D}_{\text{test}}\mathbf{Y}, \tag{1}$$

where $\hat{\mathbf{Y}} = f(\mathbf{H}, \mathbf{X}, \mathbf{D}_{\text{trn}}\mathbf{Y})$. That is, the model is trained using only the labeled nodes in $\mathbf{D}_{\text{trn}}\mathbf{Y}$, and evaluated on its ability to generalize to test nodes specified by $\mathbf{D}_{\text{test}}$. Note that $f$ is a general representation of both training-based and training-free node classifiers.

### 2.2 Related work

**Hypergraph neural networks (HNNs)**  HNNs have emerged as powerful tools for modeling higher-order relationships, where each hyperedge can connect an arbitrary number of nodes [6, 29]. This expressive capacity enables applications in diverse domains such as recommendation systems, biological networks, and knowledge graphs. Early models such as HGNN [6] extend graph convolution to the hypergraph domain via spectral approximation. UniGNN [10] provides a unified message-passing framework for both graphs and hypergraphs, while AllSet [3] introduces a permutation-invariant design that separates set encoding from message propagation. More recently, ED-HNN [23] explores the connection between the general class of hypergraph diffusion algorithms and the design of HNNs to improve expressiveness in hypergraph settings. Despite their representational power, these models often suffer from overfitting or scalability issues when training labels are scarce.

**Linear graph neural networks (GNNs)**  GNNs are highly effective for learning node representations from relational data [8, 14, 28]. However, their reliance on repeated non-linear transformations across

Table 1: Linearized forms of five representative HNNs. All parameters in each layer are integrated into a single weight matrix $\mathbf{W}$ due to the linearization, and the $l$-hop adjacency matrix $\mathbf{A}_l$ is generated from the incidence matrix $\mathbf{H}$ without involving any additional parameters.

| Method | Linearized form | Adjacency matrix |
|---|---|---|
| HGNN | $\hat{\mathbf{Y}} = \mathbf{A}_L \mathbf{X} \mathbf{W}$ | $\mathbf{A}_l = (\mathbf{D}_{\mathrm{v}}^{-\frac{1}{2}} \mathbf{H} \mathbf{D}_{\mathrm{e}}^{-1} \mathbf{H}^\top \mathbf{D}_{\mathrm{v}}^{-\frac{1}{2}})^l$ |
| HNHN | $\hat{\mathbf{Y}} = \mathbf{A}_L \mathbf{X} \mathbf{W}$ | $\mathbf{A}_l = (\mathbf{D}_{\mathrm{v},\alpha}^{-1} \mathbf{H} \mathbf{D}_{\mathrm{e}}^{\alpha} \mathbf{D}_{\mathrm{e},\beta}^{-1} \mathbf{H}^\top \mathbf{D}_{\mathrm{v}}^{\beta})^l$ |
| UniGCNII | $\hat{\mathbf{Y}} = \left( \sum_{l=0}^{L-1} \alpha(1-\alpha)^l \mathbf{A}_l + (1-\alpha)^L \mathbf{A}_L \right) \mathbf{X} \mathbf{W}$ | $\mathbf{A}_l = (\mathbf{D}_{\mathrm{v}}^{-1} \mathbf{H} \tilde{\mathbf{D}}_{\mathrm{e}}^{-1} \mathbf{H}^\top)^l$ |
| AllDeepSet | $\hat{\mathbf{Y}} = \mathbf{A}_L \mathbf{X} \mathbf{W}$ | $\mathbf{A}_l = (\mathbf{D}_{\mathrm{v}}^{-1} \mathbf{H} \mathbf{D}_{\mathrm{e}}^{-1} \mathbf{H}^\top)^l$ |
| ED-HNN | $\hat{\mathbf{Y}} = \left( \sum_{l=0}^{L-1} \alpha(1-\alpha)^l \mathbf{A}_l + (1-\alpha)^L \mathbf{A}_L \right) \mathbf{X} \mathbf{W}$ | $\mathbf{A}_l = (\mathbf{D}_{\mathrm{v}}^{-1} \mathbf{H} \mathbf{D}_{\mathrm{e}}^{-1} \mathbf{H}^\top)^l$ |

layers leads to high computational costs and limited scalability. To mitigate this, several linear GNN variants have been proposed, which simplify architecture by removing intermediate non-linearity and decoupling transformation from propagation [7, 26]. For instance, SGC [26] eliminates nonlinearities and applies a fixed propagation matrix multiple times after a single feature transformation. APPNP [7] adopts a personalized PageRank-based propagation scheme that retains a residual connection to the input features, thereby mitigating over-smoothing without increasing parameter count. $\mathrm{S}^2\mathrm{GC}$ [35] improved SGC by manually adjusting the strength of self-loops, increasing the number of propagation steps. More recent models such as SlimG [31] explore the linearized form of GNNs and further improve generalizability and interpretability. These approaches demonstrate that linear architectures can match or surpass nonlinear GNNs in performance, especially with sparse labels.

## 3 Proposed method: ZEN

We introduce ZEN, a linear hypergraph neural network (HNN) for fast, scalable, and generalizable node classification in few-shot settings. Our method builds on a unified linear formulation of existing HNNs (Section 3.1). Leveraging this formulation, we develop a closed-form approximation for the weight matrix (Section 3.2) to eliminate iterative training and propose a redundancy-aware design for the propagation matrix (Section 3.3) to mitigate structural redundancy.

### 3.1 General form of a linearized HNN

Linearization [20, 25, 26, 31] can effectively simplify the formulation of machine learning models and reduce their computational complexity, resulting in improved robustness, scalability, and interpretability. We conduct the linearization of five representative HNNs: HGNN [6], HNHN [4], UniGCNII [10], AllDeepSets [3], ED-HNN [23], and introduce their linearized forms in Table 1. We provide the full process of linearization in Appendix B. We remove all non-linear functions, including the activation functions, and integrate multiple weight matrices multiplied together into the single equivalent matrix $\mathbf{W}$ of size $d \times c$.

Then, we generalize the linearized forms of HNNs as follows:

$$\hat{\mathbf{Y}} = \left( \sum_{l=0}^{L} \alpha_l \mathbf{A}_l \right) \mathbf{X} \mathbf{W}, \tag{2}$$

where $\mathbf{A}_l$ denotes the $l$-hop adjacency matrix created from $\mathbf{H}$, and $\alpha_l$ is a hyperparameter. The matrix $\mathbf{A}_l$ determines how the incidence matrix is converted into an adjacency matrix between nodes, while $\alpha_l$ determines how much information is propagated from the $l$-hop neighbors. It is noteworthy that $\mathbf{A}_l$ does not contain any learnable parameters, since all parameters are already integrated into $\mathbf{W}$. For simplicity, we denote $\mathbf{P} = \sum_l \alpha_l \mathbf{A}_l$ and call it a *propagation matrix* in the rest of this paper.

The formulation in Eq. (2) reveals that the performance of a linear HNN is primarily determined by two factors: (a) the design of the propagation matrix $\mathbf{P}$, which aggregates multi-hop structures $\mathbf{A}_l$ with appropriate coefficients $\alpha_l$, and (b) the optimization of the weight matrix $\mathbf{W}$. We elaborate on the principled design of the propagation matrix in Section 3.3. In Section 3.2, we propose a tractable closed-form solution for $\mathbf{W}$, which eliminates the need for iterative training via backpropagation.

## 3.2 Tractable closed-form solution for the weight matrix $\mathbf{W}$

There are two approaches to obtain the optimal weight matrix for linear HNNs. The first approach is to iteratively update $\mathbf{W}$ through backpropagation until it reaches a steady state. Although it is a popular and reasonable approach, we argue that the linear characteristic of the model is not fully exploited in this way. The second approach is to derive a closed-form solution. Given training labels $\mathbf{D}_{\mathrm{trn}}\mathbf{Y}$, we directly compute the optimal $\mathbf{W}^*$ as a function of $\mathbf{D}_{\mathrm{trn}}\mathbf{Y}$ without having any iterative process. The limitation is its large computational complexity; it is rarely used in practice due to the cubic time complexity $O(d^3)$ required to compute the pseudoinverse of a matrix.

Therefore, we propose a tractable approximation of the cloesd-form solution to eliminate the need for a pseudoinverse, while maximizing its scalability by removing dependence on iterative learning. To fully linearize the objective function, we consider the squared error (SSE) loss instead of the typical cross-entropy loss which involves the non-linear softmax function. The closed-form solution of $\mathbf{W}$, without any approximation or optimization, is given as Lemma 1.

**Lemma 1.** *Given a linear HNN $\hat{\mathbf{Y}} = \mathbf{PXW}$ and the SSE loss $\mathcal{L}_{\mathrm{SSE}} = \|\mathbf{D}_{\mathrm{trn}}\hat{\mathbf{Y}} - \mathbf{D}_{\mathrm{trn}}\mathbf{Y}\|_{\mathrm{F}}^2$, the closed-form solution $\mathbf{W}^*$ that minimizes $\mathcal{L}_{\mathrm{SSE}}$ is given by*

$$\mathbf{W}^* = ((\mathbf{PX})^\top \mathbf{D}_{\mathrm{trn}}(\mathbf{PX}))^\dagger (\mathbf{PX})^\top \mathbf{D}_{\mathrm{trn}}\mathbf{Y}, \tag{3}$$

*where $\dagger$ denotes the Moore-Penrose inverse (or the pseudoinverse) of a matrix.*

*Proof.* The full proof is provided in Appendix A.1. $\qquad\square$

The problem of Eq. (3) is the pseudoinverse of matrix $(\mathbf{PX})^\top \mathbf{D}_{\mathrm{trn}}(\mathbf{PX}) \in \mathbb{R}^{d\times d}$, whose computational complexity is $O(d^3)$. To avoid the pseudoinverse by safely approximating the closed-form solution, we introduce two assumptions on the matrix $\mathbf{PX}$ which we call the *embedding matrix* of nodes before being mapped to the class space by the matrix $\mathbf{W}$.

**Assumption 1.** *Each row vector of $\mathbf{PX}$ has a unit norm, i.e., $(\mathbf{PX})_{i,:}(\mathbf{PX})_{i,:}^\top = 1$ for all $i$.*

**Assumption 2.** *For any two row vectors of $\mathbf{PX}$, their intra-class cosine similarity is approximately $1 - \epsilon$, while their inter-class cosine similarity is approximately $\epsilon$. That is, $(\mathbf{PX})_{i,:}(\mathbf{PX})_{j,:}^\top \approx 1 - \epsilon$ for $i, j$ in the same class, while $(\mathbf{PX})_{i,:}(\mathbf{PX})_{j,:}^\top \approx \epsilon$ for $i, j$ in different classes, with small $\epsilon$.*

Assumption 1 can be easily achieved by normalizing the embedding matrix $\mathbf{PX}$. Assumption 2 is harder to satisfy, but is reasonable to assume in many cases where the node feature matrix $\mathbf{X}$ provides sufficient information for classification when combined with the structural information encoded in $\mathbf{P}$, especially when we loosen the error bound $\epsilon$. Then, we propose Theorem 1 for approximating the closed-form solution of $\mathbf{W}$ with the introduced assumptions.

**Theorem 1.** *Under Assumption 1 and 2, the following holds:*

$$\mathbf{W}^* = ((\mathbf{PX})^\top \mathbf{D}_{\mathrm{trn}}(\mathbf{PX}))^\dagger (\mathbf{PX})^\top \mathbf{D}_{\mathrm{trn}}\mathbf{Y} \approx \frac{1}{\epsilon}(\mathbf{PX})^\top \mathbf{D}_{\mathrm{trn}}\mathbf{Y}.$$

*Proof.* The full proof is provided in Appendix A.2. $\qquad\square$

By Theorem 1, the burdensome pseudoinverse in the optimal weight matrix can be approximated as $(1/\epsilon)\mathbf{I}$, which is a constant matrix. Since the constant $1/\epsilon$ is multiplied to all dimensions, it does not change the result of prediction; we can safely remove it from our model.

To satisfy Assumption 1, we apply the row-wise L2 normalization for the embedding matrix $\mathbf{PX}$ as $g_{\mathrm{row}}(\mathbf{PX})$ where the function $g_{\mathrm{row}}$ denotes the row-wise normalization operator. Consequently, we normalize the weight matrix $\mathbf{W}^*$ as well, since it represents the *embedding matrix of labels;* each column of it can be understood as the $d$-dimensional embedding of each label. Without normalizing $\mathbf{W}^*$, the class with a large-norm embedding gets an advantage in the prediction stage. Therefore, we normalize both node and class embeddings, and our final method is expressed as follows:

$$\hat{\mathbf{Y}} = g_{\mathrm{row}}(\mathbf{PX})g_{\mathrm{col}}(g_{\mathrm{row}}(\mathbf{PX})^\top \mathbf{D}_{\mathrm{trn}}\mathbf{Y}). \tag{4}$$

### 3.3 Redundancy-aware propagation for eliminating self-information

A key design objective in linear HNNs is to flexibly control the influence of each $l$-hop neighborhood on node representations. This is achieved through a propagation matrix of the form

$$\mathbf{P} = \sum_{l=0}^{L} \alpha_l \mathbf{A}_l, \tag{5}$$

which denotes a weighted combination of $l$-hop adjacency matrix $\mathbf{A}_l$ multiplied with the coefficient $\alpha_l$. Since $\mathbf{PX}$ is row-normalized as shown in Eq. (4), without loss of generality, we can constrain the coefficients to lie on the probability simplex: $\sum_{l=0}^{L} \alpha_l = 1$, where $\alpha_l \geq 0$ for all $l$. By carefully controlling the coefficients for each dataset, determining how much information to take from each $l$-hop neighborhood, a linear HNN can exhibit superior performance. For example, $\alpha_1$ can be large for homophilic graphs, but small for heterophilic graphs following a typical assumption.

However, the formulation in Eq. (5) inevitably contains *redundant self-information*, which hinders the precise adjustment of neighborhood influence. We formalize this phenomenon as follows:

**Definition 1.** *Given $\mathbf{A}_l$ with $l > 0$, we define its residual self-information as the diagonal matrix*

$$\mathrm{RSI}(\mathbf{A}_l) := \mathrm{diag}(\mathbf{A}_l) \in \mathbb{R}^{n \times n}, \tag{6}$$

*which quantifies the amount of information returning to each self-node.*

The self-information of $\mathbf{A}_l$ can arise not only from the self-loops, but also from cycles or return paths in hypergraph walks if $l \geq 2$. Such self-information is not a notable problem, but even promoted in nonlinear HNNs which aim to avoid losing the information of initial node features when deep layers are adopted. However, in linear HNNs, self-information repeatedly appearing in different-hop adjacency matrices is redundant and prevents one from fully optimizing the propagation function for each dataset with a unique characteristic. Another problem is that the self-information in $\mathbf{A}_l$, if it is used for creating $\mathbf{A}_{l'}$ with $l' > l$, results in boosting the effect of local neighborhoods in $\mathbf{A}_{l'}$, making the self-information even stronger in later adjacency matrices.

To overcome these limitations, we propose a *redundancy-aware propagation* that explicitly eliminates redundant self-information from the $l$-hop adjacency matrix $\mathbf{A}_l$. Specifically, we define the $l$-hop adjacency matrix $\mathbf{A}_l^*$ without redundant self-information as follows:

$$\mathbf{A}_l^* := \hat{\mathbf{A}}_l - \mathrm{RSI}(\hat{\mathbf{A}}_l), \tag{7}$$

where $\hat{\mathbf{A}}_l := \mathbf{A}_1^* \mathbf{D}_{l-1}^* \mathbf{A}_{l-1}^*$ is the normalized adjacency matrix, $\mathbf{D}_{l-1}^*$ is the degree matrix designed specifically for the normalization method used in creating $\mathbf{A}_l$, and $\mathbf{A}_0^* := \mathbf{I}$. Then, we replace all $\mathbf{A}_l$ with $\mathbf{A}_l^*$ in Eq. (5) and use it as our propagation matrix $\mathbf{P}^* = \sum_l \alpha_l \mathbf{A}_l^*$. This ensures that the self-information in $\mathbf{P}^*$ is exactly determined by $\alpha_0$, and is not confounded by higher-hop structures. As a result, it allows a precise control over self- versus neighbor- information, which is critical in few-shot scenarios where overfitting to redundant self-signals can hinder generalization.

**Normalization of adjacency matrices**    All linearized HNNs in Table 1 normalize the adjacency matrix $\mathbf{A}_l$ either by symmetric or row normalization to make it numerically stable in deep layers. However, if we apply the same original normalization and remove self-information from normalized $\mathbf{A}_l$, it makes over-normalization and gradually diminishes feature magnitudes across layers. Thus, we re-normalize $\mathbf{A}_l$ and derive the following matrices for $l = 1, 2$:

$$\hat{\mathbf{A}}_1 = \mathbf{D}_{\mathrm{v}}^{-\frac{1}{2}} \mathbf{H} (\mathbf{D}_{\mathrm{e}} - \mathbf{I})^{-1} \mathbf{H}^\top \mathbf{D}_{\mathrm{v}}^{-\frac{1}{2}} \tag{8}$$

$$\hat{\mathbf{A}}_2 = \mathbf{A}_1^* \left( \mathbf{D}_{\mathrm{v}}^{\frac{1}{2}} (\mathbf{D}_{\mathrm{v}} - \mathbf{I})^{-1} \mathbf{D}_{\mathrm{v}}^{\frac{1}{2}} \right) \mathbf{A}_1^*. \tag{9}$$

**Avoiding dense adjacency matrices**    The main challenge for Eq. (7) is deriving the dense matrix $\hat{\mathbf{A}}_l$ of size $|\mathcal{V}| \times |\mathcal{V}|$ for the computation of its self-information. We provide explicit expressions of the RSI terms for $l = 1$ and $l = 2$, since we set $L$ to 2 in our experiments. Our derivation is based on symmetric normalization, but the framework is readily extensible to row normalization. Detailed derivations for the row-normalized variant are included in Appendix D.

**Lemma 2.** *Given $\hat{\mathbf{A}}_1$ in Eq. (8), $\mathrm{RSI}(\hat{\mathbf{A}}_1)$ is given by*

$$(\mathrm{RSI}(\hat{\mathbf{A}}_1))_{ii} = d_{v_i}^{-1} \left( \sum_{e_j \in \mathcal{N}(v_i)} (d_{e_j} - 1)^{-1} \right), \tag{10}$$

*where $d_x$ denotes the degree of node $x$ or the number of nodes in hyperedge $x$, based on the type of $x$, and $\mathcal{N}(v_i)$ denotes the set of hyperedges incident to node $v_i$.*

*Proof.* The full proof is provided in Appendix A.3. □

**Lemma 3.** *Given $\mathbf{A}_1^* = \hat{\mathbf{A}}_1 - \mathrm{RSI}(\hat{\mathbf{A}}_1)$ and $\hat{\mathbf{A}}_2$ in Eq. (9), $\mathrm{RSI}(\hat{\mathbf{A}}_2)$ is given by*

$$(\mathrm{RSI}(\hat{\mathbf{A}}_2))_{ii} = d_{v_i}^{-1} \left( \sum_{e_j \in \mathcal{N}(v_i)} (d_{e_j} - 1)^{-2} \left( \sum_{v_k \in \mathcal{N}(e_j) \setminus \{v_i\}} (d_{v_k} - 1)^{-1} \right) \right), \tag{11}$$

*where $d_x$ denotes the degree of node $x$ or the number of nodes in hyperedge $x$, based on the type of $x$, and $\mathcal{N}(v_i)$ denotes the set of hyperedges incident to node $v_i$.*

*Proof.* The proof follows by applying the same reasoning as in Lemma 2. □

From the 3-hop neighborhood onward, self-information can return through cycles rather than simple backtracking paths, complicating both its identification and principled removal during propagation. Additionally, deeper propagation exacerbates the computational burden and enlarges the hyperparameter space. For these reasons, we restrict our model to 2-hop propagation, which strikes a balance between expressive power and efficiency. This design choice is supported by two key observations: (a) Increasing the number of hops introduces a linearly growing number of propagation coefficients $\alpha_l$, resulting in the quadratic expansion of the hyperparameter space. (b) Empirically, most HNNs attain competitive performance with only 1–2 propagation layers.

### 3.4 Summary of ZEN and its computational complexity

Putting it all together, the ZEN classifier $f^*$ can be summarized as follow:

$$f^*(\mathbf{H}, \mathbf{X}, \mathbf{D}_{\mathrm{trn}}\mathbf{Y}) = g_{\mathrm{row}}(\mathbf{P}^*\mathbf{X}) g_{\mathrm{col}}(g_{\mathrm{row}}(\mathbf{P}^*\mathbf{X})^\top \mathbf{D}_{\mathrm{trn}}\mathbf{Y}), \tag{12}$$

where $\mathbf{P}^* = \alpha_0 \mathbf{A}_0^* + \alpha_1 \mathbf{A}_1^* + \alpha_2 \mathbf{A}_2^*$ with three hyperparameters $\alpha_0$, $\alpha_1$, and $\alpha_2$, and $\mathbf{A}_l^*$ denotes the refined $l$-hop adjacency matrix which eliminates the redundant self-information.

The function $f^*$ is a closed-form prediction formula that directly produces label predictions for all nodes given a small set of labeled nodes. Crucially, it eliminates the need for iterative training: the labels of the training nodes are injected as explicit inputs rather than being implicitly encoded via gradient-based optimization. This design enables extremely fast and stable inference, particularly under low-resource scenarios such as few-shot settings, where traditional training-based models often suffer from instability due to limited supervision. More importantly, by removing self-information explicitly, ZEN avoids redundancy and enables fully optimized combinations of multi-hop structure within the probability simplex. This capability is fundamentally absent in conventional propagation schemes, where self-information is entangled with higher-hop structures.

The computational complexity of $f^*$ depends on the two main stages. The first is the dense matrix multiplication between $g_{\mathrm{row}}(\mathbf{P}^*\mathbf{X})$ and $g_{\mathrm{col}}(g_{\mathrm{row}}(\mathbf{P}^*\mathbf{X})^\top \mathbf{D}_{\mathrm{trn}}\mathbf{Y})$, which is $O(|\mathcal{V}|dc)$, linear in the number of nodes, feature dimension, and number of classes. The second is the construction of $\mathbf{P}^*$, where we remove self-information from each $\mathbf{A}_l$ for all $l \geq 1$. This has the same time complexity as standard message passing schemes, $O(\mathrm{nnz}(\mathbf{H}) \cdot dL)$, where $\mathrm{nnz}(\mathbf{H})$ denotes the number of non-zero entries in the hypergraph structure, and the number $L$ of layers is set to 2 in our experiments.

## 4 Experiments

We conduct comprehensive experiments on 11 real-world hypergraphs to verify the effectiveness of ZEN. We show that ZEN consistently outperforms existing HNNs in few-shot node classification tasks while exhibiting remarkable scalability. We then present a case study highlighting the interpretability of ZEN, which comes from its linear decision mechanism, on a real hypergraph.

Table 2: Statistics of datasets. The first ten datasets are used as the main benchmark for evaluating accuracy, while the Zoo dataset is used for interpretability analysis.

| | Cora | Citeseer | Pubmed | Cora-CA | 20News | MN40 | Congress | Walmart | Senate | House | Zoo |
|---|---|---|---|---|---|---|---|---|---|---|---|
| # nodes $|\mathcal{V}|$ | 2,708 | 3,312 | 19,717 | 2,708 | 16,242 | 12,311 | 1,718 | 88,860 | 282 | 1,290 | 101 |
| # edges $|\mathcal{E}|$ | 1,579 | 1,079 | 7,963 | 1,072 | 100 | 12,311 | 83,105 | 69,906 | 315 | 340 | 43 |
| # classes $c$ | 7 | 6 | 3 | 7 | 4 | 40 | 2 | 11 | 2 | 2 | 7 |
| # features $d$ | 1,433 | 3,703 | 500 | 1,433 | 100 | 100 | 100 | 100 | 100 | 100 | 16 |
| density ($\frac{|\mathcal{E}|}{|\mathcal{V}|}$) | 0.5835 | 0.3258 | 0.4041 | 0.3959 | 0.0062 | 1.0000 | 48.3946 | 0.7868 | 1.1160 | 0.2636 | 0.4257 |

Table 3: Classification accuracy (%) for 5-shot node classification on real-world hypergraphs. We report the mean and standard deviation over 10 runs. Boldfaced letters indicate the best accuracy, and underlined letters indicate the second. ZEN achieves the highest average rank.

| Methods | Cora | Citeseer | Pubmed | Cora_CA | 20News | MN40 | Congress | Walmart | Senate | House | Avg. Rank |
|---|---|---|---|---|---|---|---|---|---|---|---|
| HGNN | $44.4_{\pm 8.9}$ | $40.1_{\pm 6.5}$ | $52.5_{\pm 9.1}$ | $54.3_{\pm 3.6}$ | $\mathbf{73.1_{\pm 2.3}}$ | $94.7_{\pm 0.3}$ | $86.7_{\pm 1.1}$ | $39.6_{\pm 2.4}$ | $56.8_{\pm 5.0}$ | $63.4_{\pm 4.3}$ | 5.9 |
| HNHN | $36.7_{\pm 5.8}$ | $36.0_{\pm 3.7}$ | $51.8_{\pm 3.7}$ | $39.2_{\pm 5.2}$ | $41.2_{\pm 5.7}$ | $90.8_{\pm 1.4}$ | $51.1_{\pm 2.7}$ | $15.9_{\pm 3.0}$ | $69.7_{\pm 11.6}$ | $67.4_{\pm 8.3}$ | 7.9 |
| HCHA | $44.4_{\pm 8.7}$ | $41.2_{\pm 6.5}$ | $52.9_{\pm 10.4}$ | $54.5_{\pm 4.2}$ | $\underline{72.9_{\pm 2.5}}$ | $94.7_{\pm 0.2}$ | $86.6_{\pm 1.3}$ | $39.3_{\pm 2.5}$ | $53.0_{\pm 5.0}$ | $63.5_{\pm 4.6}$ | 5.9 |
| UniGCN | $48.5_{\pm 8.3}$ | $41.6_{\pm 3.7}$ | $54.2_{\pm 10.3}$ | $55.3_{\pm 4.3}$ | $70.4_{\pm 2.8}$ | $95.9_{\pm 0.3}$ | $\mathbf{91.6_{\pm 0.4}}$ | $40.1_{\pm 2.8}$ | $61.4_{\pm 4.4}$ | $67.9_{\pm 5.1}$ | 3.9 |
| UniGCNII | $43.3_{\pm 9.9}$ | $38.9_{\pm 6.7}$ | $54.5_{\pm 8.4}$ | $52.0_{\pm 4.5}$ | $66.5_{\pm 4.6}$ | $\underline{96.4_{\pm 0.4}}$ | $83.5_{\pm 6.4}$ | $23.5_{\pm 2.4}$ | $\mathbf{70.4_{\pm 8.5}}$ | $\underline{70.7_{\pm 7.4}}$ | 5.5 |
| AllDeepSets | $48.6_{\pm 4.7}$ | $42.6_{\pm 4.4}$ | $53.2_{\pm 5.8}$ | $55.3_{\pm 5.1}$ | $51.4_{\pm 4.4}$ | $94.7_{\pm 0.3}$ | $69.5_{\pm 4.7}$ | $24.5_{\pm 3.7}$ | $65.3_{\pm 10.3}$ | $63.4_{\pm 8.3}$ | 5.7 |
| AllSetTransformer | $\underline{50.5_{\pm 4.4}}$ | $\underline{44.8_{\pm 2.7}}$ | $60.4_{\pm 4.5}$ | $59.6_{\pm 3.4}$ | $70.3_{\pm 1.5}$ | $95.5_{\pm 0.2}$ | $88.2_{\pm 1.1}$ | $38.3_{\pm 6.4}$ | $63.1_{\pm 10.7}$ | $66.3_{\pm 8.3}$ | $\underline{3.6}$ |
| ED-HNN | $48.4_{\pm 6.4}$ | $44.5_{\pm 3.5}$ | $56.5_{\pm 6.6}$ | $58.8_{\pm 3.8}$ | $67.7_{\pm 3.7}$ | $96.0_{\pm 0.2}$ | $\underline{89.1_{\pm 4.0}}$ | $\underline{42.9_{\pm 5.7}}$ | $63.1_{\pm 9.1}$ | $62.8_{\pm 10.4}$ | 4.1 |
| **ZEN (proposed)** | $\mathbf{51.9_{\pm 10.1}}$ | $\mathbf{49.1_{\pm 4.8}}$ | $\mathbf{62.6_{\pm 3.9}}$ | $\mathbf{60.0_{\pm 6.2}}$ | $68.6_{\pm 4.8}$ | $\mathbf{97.6_{\pm 0.3}}$ | $87.0_{\pm 4.8}$ | $\mathbf{43.9_{\pm 3.1}}$ | $\mathbf{70.4_{\pm 10.0}}$ | $\mathbf{73.2_{\pm 6.3}}$ | **1.7** |

## 4.1 Experimental setup

**Datasets.** We evaluate ZEN on a total of 11 real-world hypergraph graphs. To assess predictive performance and computational efficiency, we use 10 standard benchmarks: Cora, Citeseer, Pubmed, Cora_CA, 20News100, ModelNet40, Congress, Walmart, Senate, and House, following prior work [15, 23]. For interpretability analysis, we use Zoo [15], a small dataset whose feature attributes are semantically interpretable. Detailed dataset statistics are provided in Table 2.

**Baselines and hyperparameters.** We compare ZEN with 8 representative hypergraph neural networks (HNN) models: HGNN [6], HNHN [4], HCHA [2], UniGCN, UniGCNII [10], AllDeepSets, AllSetTransformer [3], and ED-HNN [23]. All baselines are implemented based on the official codebase of ED-HNN, which provides a unified framework for fair comparison. All baseline are trained using the Adam optimizer with no weight decay, and we conduct a grid search over 72 hyperparameter configurations: $\text{lr} \in \{10^{-2}, 10^{-3}, 10^{-4}\}$, $\text{epochs} \in \{50, 100, 150, 200\}$, $\text{num\_layers} \in \{1, 2\}$, $\text{hidden\_dim} \in \{64, 128, 256\}$. In contrast, ZEN requires no training hyperparameters. Instead, we search over 55 combinations of propagation coefficients $(\alpha_0, \alpha_1, \alpha_2)$ uniformly sampled from the 2-simplex, yielding a comparable hyperparameter space size to that of baselines. For each dataset split, we report the test accuracy corresponding to the best validation performance. All our experiments are conducted with NVIDIA RTX A6000 and AMD EPYC 9354.

**Evaluation.** We evaluate the accuracy of all models on 10 random data splits per dataset. For each split, we allocate 5 labeled nodes per class for training, and another 5 nodes per class for validation [12, 27], making 5-shot node classification. The remaining nodes are used for testing. We report the average classification accuracy and standard deviation across the ten splits [11, 17, 19].

## 4.2 Classification accuracy

Table 3 compares the accuracy of ZEN and the baseline HNNs on 10 hypergraphs. ZEN demonstrates competitive or superior performance across all datasets, showing the highest average rank. Despite its simple linear architecture, ZEN achieves high accuracy even on complex hypergraph structures, being competitive with complicated nonlinear methods. This validates the effectiveness of ZEN's architectural design in capturing high-order relationships, and highlights its strong generalization ability in few-shot node classification scenarios, where model robustness is crucial.

The results also highlight intriguing trends among baseline models. In particular, early models such as HGNN and UniGCN remain competitive, particularly on datasets such as 20News and Congress. Their relatively simple architectures may contribute to stronger generalization in few-shot settings,

Table 4: The running time of ZEN and the baseline models, including both training and inference. Each time is represented as a ratio over the running time of ZEN. Therefore, the lower is the better. ZEN consistently shows the fastest runtime, up to $696\times$ faster than the best competitor.

| Methods | Cora | Citeseer | Pubmed | Cora_CA | 20News | MN40 | Congress | Wallmart | Senate | House |
|---|---|---|---|---|---|---|---|---|---|---|
| HGNN | 8.65 | 3.81 | 3.00 | 8.95 | 10.54 | 20.05 | 53.10 | 30.16 | 777.14 | 388.05 |
| HNHN | 7.61 | 2.71 | 3.56 | 7.23 | 15.51 | 13.78 | 42.11 | 21.47 | 696.85 | 345.93 |
| HCHA | 12.37 | 4.74 | 5.51 | 10.19 | 12.21 | 20.63 | 71.62 | 12.73 | 1008.85 | 699.41 |
| UniGCN | 17.12 | 2.63 | 8.29 | 8.97 | 28.79 | 21.86 | 91.16 | 32.90 | 716.43 | 292.95 |
| UniGCNII | 15.77 | 5.18 | 2.56 | 17.25 | 16.98 | 19.41 | 80.17 | 36.37 | 696.63 | 369.10 |
| AllDeepSets | 58.11 | 24.92 | 11.01 | 57.56 | 35.52 | 47.51 | 273.58 | 90.16 | 4048.55 | 1748.67 |
| AllSetTransformer | 9.76 | 4.30 | 10.98 | 12.37 | 60.21 | 24.94 | 65.59 | 76.89 | 997.43 | 524.15 |
| ED-HNN | 16.04 | 6.06 | 5.46 | 23.55 | 46.71 | 28.70 | 426.91 | 46.31 | 714.73 | 379.41 |
| ZEN (proposed) | 1.00 | 1.00 | 1.00 | 1.00 | 1.00 | 1.00 | 1.00 | 1.00 | 1.00 | 1.00 |

where overfitting is a common challenge. In contrast, more complex methods such as HNHN and AllDeepSets tend to underperform, likely due to higher complexity and reduced robustness under limited supervision. These observations further underscore the strength of the simple yet effective design of ZEN, producing consistently high performance in diverse hypergraph structures.

## 4.3 Running time

Table 4 shows the running time of ZEN and the baseline models, including both training and inference time. All existing HNNs exhibit significantly higher computational costs over ZEN. The speedup stands out for large complicated HNNs, such as AllDeepSets and ED-HNN, as ZEN is over $1700\times$ faster than AllDeepSets in the House dataset. Even for relatively simple models such as HGNN, HCHA, UniGCN, and UniGCNII, ZEN shows a consistent improvement ranging from $2.5\times$ even to $300\times$. In summary, ZEN demonstrates overwhelming speed superiority across all datasets while maintaining competitive accuracy, establishing itself as a highly efficient solution.

The efficiency of ZEN comes from its lightweight architecture, where the computational cost is linear in the number of nodes, feature dimension and number of classes. This design leads to substantial runtime advantages on datasets with compact input dimensions. In particular, Congress, Senate, and House have feature dimensions that are at least $5\times$ and up to $37\times$ smaller, node counts up to $9\times$ fewer, and class counts between $2\times$ and $20\times$ fewer than other datasets. These characteristics make them ideal for highlighting the scalability of ZEN, which achieves remarkable speedups of up to $292\times$ on House and $696\times$ on Senate while maintaining competitive accuracy.

## 4.4 Interpretability

ZEN is inherently interpretable, thanks to its linear decision process that directly maps the feature space to the prediction. There are two ways to interpret the learned knowledge of ZEN. First, the $(i, k)$-th element of the weight matrix $\mathbf{W}^*$ can be understood as the importance of the $i$-th feature for predicting the $k$-th class. Second, each column of $\mathbf{W}^*$ can be understood as the embedding of class $k$ lying on the feature space. In this way, the relationships between class embeddings and node features provide a deeper insight on the nature of the given dataset, along with the graphical structure.

To verify the interpretability of ZEN, we conduct a case study on the Zoo dataset, whose node features have clear semantic meanings: each feature attribute represents a characteristic of an animal, e.g., hair or milk, while the target class is its species. The nodes represent animals, and the hyperedges group together animals that share a common feature, e.g., all animals having the same *hair*.

Table 5 visualizes the weight matrix $\mathbf{W}^*$, where each value is color-coded: darker red indicates a higher relative value compared to other classes for that feature. Since the initial input features are all nonnegative, the resulting weight elements also remain nonnegative. For instance, the Mammal class shows significantly higher values in Hair, Milk, and Capsize, suggesting that these features play a key role in distinguishing mammals from other animal groups. Similarly, the Bird class exhibits prominent values in Feathers, Eggs, Airborne, etc., reflecting biologically distinctive traits of birds. These results demonstrate that ZEN not only achieves high predictive performance but also yields representations that align with domain knowledge in a transparent and interpretable manner.

Table 5: Relative feature importance values learned by ZEN on the Zoo dataset under a 3-shot setting. Darker red cells indicate higher values relative to other classes, with cells having darkness of 80% or higher highlighted by black boxes. Refer to Section 4.4 for detailed information.

| | Mammal | Bird | Reptile | Fish | Amphibian | Bug | Invertebrate |
|---|---|---|---|---|---|---|---|
| Hair | 0.1735 | 0.0719 | 0.0778 | 0.0819 | 0.0661 | 0.0800 | 0.0465 |
| Feathers | 0.0251 | 0.1726 | 0.0383 | 0.0414 | 0.0311 | 0.0305 | 0.0276 |
| Eggs | 0.1072 | 0.2440 | 0.2255 | 0.2931 | 0.1979 | 0.1792 | 0.1850 |
| Milk | 0.1720 | 0.0664 | 0.0740 | 0.0826 | 0.0626 | 0.0432 | 0.0409 |
| Airborne | 0.0310 | 0.1417 | 0.0450 | 0.0456 | 0.0376 | 0.0979 | 0.0361 |
| Aquatic | 0.0807 | 0.0688 | 0.0727 | 0.2522 | 0.1606 | 0.0492 | 0.1437 |
| Predator | 0.1796 | 0.1054 | 0.1878 | 0.2841 | 0.1601 | 0.1006 | 0.1688 |
| Toothed | 0.1707 | 0.1001 | 0.1946 | 0.2921 | 0.1959 | 0.0682 | 0.0745 |
| Backbone | 0.2280 | 0.2768 | 0.2687 | 0.3380 | 0.2304 | 0.1012 | 0.1049 |
| Breathes | 0.2239 | 0.2769 | 0.2597 | 0.1652 | 0.2235 | 0.1939 | 0.1064 |
| Venomous | 0.0109 | 0.0141 | 0.0628 | 0.0217 | 0.0465 | 0.0415 | 0.171 |
| Fins | 0.0246 | 0.0272 | 0.0337 | 0.2027 | 0.0301 | 0.0183 | 0.0273 |
| Legs | 0.8380 | 0.7794 | 0.7956 | 0.5875 | 0.8520 | 0.9333 | 0.9310 |
| Tail | 0.1836 | 0.2642 | 0.2542 | 0.3204 | 0.1522 | 0.0913 | 0.0963 |
| Domestic | 0.0225 | 0.0266 | 0.0234 | 0.0253 | 0.0197 | 0.0179 | 0.0146 |
| Capsize | 0.1416 | 0.1515 | 0.1134 | 0.1887 | 0.0687 | 0.0487 | 0.0744 |

Table 6: Ablation study of ZEN with four baselines: three variants with a selective removal of the ZEN components, and the linearized HGNN that lacks multi-hop message combination. Our full model consistently outperforms all ablations, demonstrating the effectiveness of both components.

| Methods | Cora | Citeseer | Pubmed | Cora_CA | 20News | MN40 | Congress | Wallmart | Senate | House | Avg Rank |
|---|---|---|---|---|---|---|---|---|---|---|---|
| Linearized HGNN | $42.7_{\pm 8.4}$ | $34.3_{\pm 9.4}$ | $51.7_{\pm 6.6}$ | $49.6_{\pm 6.9}$ | $68.0_{\pm 6.0}$ | $94.5_{\pm 0.5}$ | $83.7_{\pm 5.0}$ | $29.0_{\pm 6.4}$ | $55.8_{\pm 5.1}$ | $57.8_{\pm 6.5}$ | 4.5 |
| No TCS, No RAP | $44.3_{\pm 7.2}$ | $35.0_{\pm 7.4}$ | $52.8_{\pm 6.0}$ | $48.8_{\pm 6.8}$ | $64.6_{\pm 7.5}$ | $97.6_{\pm 0.5}$ | $\mathbf{88.5_{\pm 2.3}}$ | $22.3_{\pm 6.6}$ | $\underline{71.8_{\pm 5.1}}$ | $69.0_{\pm 4.6}$ | 3.7 |
| No TCS | $46.9_{\pm 5.5}$ | $37.8_{\pm 6.9}$ | $53.3_{\pm 6.0}$ | $52.5_{\pm 5.2}$ | $\mathbf{69.8_{\pm 7.5}}$ | $\mathbf{97.8_{\pm 0.2}}$ | $87.0_{\pm 2.5}$ | $26.6_{\pm 5.0}$ | $67.2_{\pm 10.1}$ | $71.6_{\pm 5.9}$ | 2.7 |
| No RAP | $50.6_{\pm 8.8}$ | $48.6_{\pm 4.4}$ | $\mathbf{62.6_{\pm 4.2}}$ | $60.0_{\pm 5.8}$ | $64.7_{\pm 4.9}$ | $97.7_{\pm 0.3}$ | $\underline{88.4_{\pm 2.5}}$ | $40.6_{\pm 4.9}$ | $\mathbf{73.8_{\pm 5.2}}$ | $71.4_{\pm 3.9}$ | $\underline{2.0}$ |
| **ZEN (proposed)** | $\mathbf{51.9_{\pm 10.1}}$ | $\mathbf{49.1_{\pm 4.8}}$ | $\mathbf{62.6_{\pm 3.9}}$ | $\mathbf{60.0_{\pm 6.2}}$ | $\underline{68.6_{\pm 4.8}}$ | $97.6_{\pm 0.3}$ | $87.0_{\pm 4.8}$ | $\mathbf{43.9_{\pm 3.1}}$ | $70.4_{\pm 10.0}$ | $\mathbf{73.2_{\pm 6.3}}$ | **1.7** |

## 4.5 Ablation study

In Table 6, we conduct an ablation study to assess the individual and combined impact of *tractable closed-form solution* (TCS) and *redundancy-aware propagation* (RAP), two core modules of ZEN, in the same setting as in Table 3. The results show that the removal of both components leads to notable performance drops, while isolating TCS or RAP yields moderate gains, with TCS generally offering more stable improvements. Our full model ZEN consistently achieves the best results, outperforming all ablated variants with the highest average rank. These findings highlight the complementary roles of TCS and RAP—TCS provides stability and tractability through a closed-form solution, while RAP enhances representation by mitigating redundancy.

We observe that removing RAP or TCS can improve performance on certain datasets (Congress and Senate for RAP, 20News and MN40 for TCS). RAP tends to degrade performance on high-density datasets such as Congress and Senate, likely because the degree normalization reduces each node's relative contribution, limiting the effect of self-information. For TCS, which assumes informative node embeddings, the impact of removal is more significant on 20News than MN40. The extremely low density of 20News may hinder ZEN, which relies solely on propagation to refine embeddings, from producing sufficiently informative representations. In contrast, MN40's large number of classes increases the relative Frobenius norm error of TCS, suggesting that the approximation becomes less accurate as the number of classes grows, partially explaining the observed results.

## 5 Conclusion

In this work, we propose ZEN (Zero-Parameter Hypergraph Neural Network), a parameter-free model for few-shot node classification on hypergraphs. By reformulating existing HNNs into a unified

linear framework with a tractable closed-form weight solution and redundancy-aware propagation, ZEN achieves strong generalization, fast inference, and interpretable representations without iterative training. Extensive experiments demonstrate its superior accuracy and scalability. One limitation of our work is that ZEN is specifically designed for node classification and is not tailored to other hypergraph tasks such as hyperedge prediction, local clustering, or hyperedge disambiguation. In future work, we plan to extend our framework to support a broader range of hypergraph learning tasks by designing a more general-purpose and efficient linear HNN architecture.

## Acknowledgements

This work was supported by the National Research Foundation of Korea (NRF) grant funded by the Korea government (MSIT) (RS-2024-00341425 and RS-2024-00406985). This work was supported by BK21 FOUR (Connected AI Education & Research Program for Industry and Society Innovation, KAIST EE, No. 4120200113769). Jaemin Yoo is the corresponding author.

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

# A Proofs

## A.1 Proof of lemma 1

$$\frac{\partial \mathcal{L}_{\text{SSE}}}{\partial \mathbf{W}} = -2(\mathbf{D}_{\text{trn}}\mathbf{PX})^{\top}(\mathbf{D}_{\text{trn}}\mathbf{Y} - \mathbf{D}_{\text{trn}}\hat{\mathbf{Y}}) \tag{13}$$

where $\hat{\mathbf{Y}} = \mathbf{PXW}$. The optimal $\mathbf{W}_*$ minimizes $\mathcal{L}_{\text{SSE}}$:

$$-2(\mathbf{D}_{\text{trn}}\mathbf{PX})^{\top}(\mathbf{D}_{\text{trn}}\mathbf{Y} - \mathbf{D}_{\text{trn}}\mathbf{PXW}_*) = 0 \tag{14}$$

$$(\mathbf{D}_{\text{trn}}\mathbf{PX})^{\top}\mathbf{D}_{\text{trn}}\mathbf{PXW}_* = (\mathbf{D}_{\text{trn}}\mathbf{PX})^{\top}\mathbf{D}_{\text{trn}}\mathbf{Y} \tag{15}$$

$$\left((\mathbf{PX})^{\top}\mathbf{D}_{\text{trn}}(\mathbf{PX})\right)\mathbf{W}_* = (\mathbf{PX})^{\top}\mathbf{D}_{\text{trn}}\mathbf{Y} \tag{16}$$

$$\mathbf{W}_* = \left((\mathbf{PX})^{\top}\mathbf{D}_{\text{trn}}(\mathbf{PX})\right)^{\dagger}(\mathbf{PX})^{\top}\mathbf{D}_{\text{trn}}\mathbf{Y} \tag{17}$$

## A.2 Proof of theorem 1

Without loss of generality, we can set the first $k \cdot c$ diagonal elements in $\mathbf{D}_{\text{trn}}$ to be 1, and they are ordered with their labels. Let $\mathbf{D}'_{\text{trn}} = [\mathbf{I}_{kc}\ \mathbf{0}] \in \mathbb{R}^{kc \times |\mathcal{V}|}$. We can easily accept $\mathbf{D}'^{\top}_{\text{trn}}\mathbf{D}'_{\text{trn}} = \mathbf{D}_{\text{trn}} = (\mathbf{D}_{\text{trn}})^2$ by definition.

We first prove the following four lemmas for proving the main theorem:

**Lemma 4.** *By definition, the following holds:*

$$\mathbf{D}'^{\top}_{\text{trn}}\mathbf{D}'_{\text{trn}} = \mathbf{D}_{\text{trn}} = (\mathbf{D}_{\text{trn}})^2 \tag{18}$$

*Proof.* The proof is straightforward. $\qquad\square$

**Lemma 5.** *Under the assumptions, $(\mathbf{D}'_{\text{trn}}(\mathbf{PX}))(\mathbf{D}'_{\text{trn}}(\mathbf{PX}))^{\top}$ can be expressed as:*

$$(\mathbf{D}'_{\text{trn}}(\mathbf{PX}))(\mathbf{D}'_{\text{trn}}(\mathbf{PX}))^{\top} = (1 - 2\epsilon)(\mathbf{I}_c \otimes \mathbf{J}_k) + \epsilon(\mathbf{I}_{kc} + \mathbf{J}_{kc}) \tag{19}$$

*where $\mathbf{J}$ is an all-one matrix, i.e. $\mathbf{J}_{kc} = \mathbf{1}_{kc}\mathbf{1}_{kc}^{\top}$, and $\otimes$ denotes Kronecker product.*

*Proof.* The proof is straightforward. $\qquad\square$

**Lemma 6.** *When $\epsilon > 0$, $(1 - 2\epsilon)(\mathbf{I}_c \otimes \mathbf{J}_k) + \epsilon(\mathbf{I}_{kc} + \mathbf{J}_{kc})$ has inverse matrix as follows:*

$$((1 - 2\epsilon)(\mathbf{I}_c \otimes \mathbf{J}_k) + \epsilon(\mathbf{I}_{kc} + \mathbf{J}_{kc}))^{-1} = \frac{1}{\lambda_1}\mathbf{M}_1 + \frac{1}{\lambda_2}\mathbf{M}_2 + \frac{1}{\lambda_3}\mathbf{M}_3 \tag{20}$$

*where $\lambda_1 = \epsilon, \lambda_2 = (1 - 2\epsilon)k + \epsilon, \lambda_3 = k(1 - 2\epsilon + \epsilon c) + \epsilon, \mathbf{M}_1 = \mathbf{I}_{kc} - \frac{1}{k}(\mathbf{I}_c \otimes \mathbf{J}_k), \mathbf{M}_2 = \frac{1}{k}(\mathbf{I}_c \otimes \mathbf{J}_k) - \frac{1}{kc}\mathbf{J}_{kc}, \mathbf{M}_3 = \frac{1}{kc}\mathbf{J}_{kc}$*

*Proof.* We consider the matrix $\mathbf{M} = (1 - 2\epsilon)(\mathbf{I}_c \otimes \mathbf{J}_k) + \epsilon(\mathbf{I}_{kc} + \mathbf{J}_{kc})$. We define three mutually orthogonal projection matrices:

$$\mathbf{M}_1 = \mathbf{I}_{kc} - \frac{1}{k}(\mathbf{I}_c \otimes \mathbf{J}_k), \quad \mathbf{M}_2 = \frac{1}{k}(\mathbf{I}_c \otimes \mathbf{J}_k) - \frac{1}{kc}\mathbf{J}_{kc}, \quad \mathbf{M}_3 = \frac{1}{kc}\mathbf{J}_{kc} \tag{21}$$

It is easily verified that these satisfy

$$\mathbf{M}_i^2 = \mathbf{M}_i, \quad \mathbf{M}_i\mathbf{M}_j = 0\ (i \neq j), \quad \mathbf{M}_1 + \mathbf{M}_2 + \mathbf{M}_3 = \mathbf{I}_{kc}. \tag{22}$$

We now compute the action of $\mathbf{M}$ on each subspace:

$$\mathbf{MM}_1 = \varepsilon\mathbf{M}_1,$$
$$\mathbf{MM}_2 = ((1 - 2\varepsilon)k + \varepsilon)\mathbf{M}_2,$$
$$\mathbf{MM}_3 = (k(1 - 2\varepsilon + \varepsilon c) + \varepsilon)\mathbf{M}_3.$$

Thus, $\mathbf{M}$ admits the spectral decomposition

$$\mathbf{M} = \lambda_1\mathbf{M}_1 + \lambda_2\mathbf{M}_2 + \lambda_3\mathbf{M}_3, \tag{23}$$

where
$$\lambda_1 = \epsilon, \quad \lambda_2 = (1 - 2\epsilon)k + \epsilon, \quad \lambda_3 = k(1 - 2\epsilon + \epsilon c) + \epsilon. \tag{24}$$
Since $\epsilon > 0$, all eigenvalues are strictly positive, so $\mathbf{M}$ is invertible. The inverse is given by
$$\mathbf{M}^{-1} = \lambda_1^{-1}\mathbf{M}_1 + \lambda_2^{-1}\mathbf{M}_2 + \lambda_3^{-1}\mathbf{M}_3. \tag{25}$$
$\square$

**Lemma 7.** *Under the small $\epsilon \ll 1$, the following holds:*
$$((\mathbf{D}'_{\mathrm{trn}}(\mathbf{PX}))(\mathbf{D}'_{\mathrm{trn}}(\mathbf{PX}))^\top)^{-2} \approx \frac{1}{\epsilon}((\mathbf{D}'_{\mathrm{trn}}(\mathbf{PX}))(\mathbf{D}'_{\mathrm{trn}}(\mathbf{PX}))^\top)^{-1} \tag{26}$$

*Proof.*
$$((\mathbf{D}'_{\mathrm{trn}}(\mathbf{PX}))(\mathbf{D}'_{\mathrm{trn}}(\mathbf{PX}))^\top)^{-1} = \frac{1}{\lambda_1}\mathbf{M}_1 + \frac{1}{\lambda_2}\mathbf{M}_2 + \frac{1}{\lambda_3}\mathbf{M}_3 \tag{27}$$
$$((\mathbf{D}'_{\mathrm{trn}}(\mathbf{PX}))(\mathbf{D}'_{\mathrm{trn}}(\mathbf{PX}))^\top)^{-2} = \frac{1}{\lambda_1^2}\mathbf{M}_1 + \frac{1}{\lambda_2^2}\mathbf{M}_2 + \frac{1}{\lambda_3^2}\mathbf{M}_3 \tag{28}$$
When $\epsilon \ll 1$, $\lambda_1 = \epsilon, \lambda_2 \approx k, \lambda_3 \approx k$. Therefore,
$$((\mathbf{D}'_{\mathrm{trn}}(\mathbf{PX}))(\mathbf{D}'_{\mathrm{trn}}(\mathbf{PX}))^\top)^{-1} \approx \frac{1}{\epsilon}\mathbf{M}_1 \tag{29}$$
$$((\mathbf{D}'_{\mathrm{trn}}(\mathbf{PX}))(\mathbf{D}'_{\mathrm{trn}}(\mathbf{PX}))^\top)^{-2} \approx \frac{1}{\epsilon^2}\mathbf{M}_1 \tag{30}$$
Thus, $((\mathbf{D}'_{\mathrm{trn}}(\mathbf{PX}))(\mathbf{D}'_{\mathrm{trn}}(\mathbf{PX}))^\top)^{-2} \approx \frac{1}{\epsilon}((\mathbf{D}'_{\mathrm{trn}}(\mathbf{PX}))(\mathbf{D}'_{\mathrm{trn}}(\mathbf{PX}))^\top)^{-1}$ with small $\epsilon \ll 1$. $\square$

The rest of this section proves the theorem based on above lemmas. First, we can reformulate $\mathbf{K}^\dagger = ((\mathbf{PX})^\top \mathbf{D}_{\mathrm{trn}}(\mathbf{PX}))^\dagger$ as:
$$\mathbf{K}^\dagger = ((\mathbf{PX})^\top \mathbf{D}'^\top_{\mathrm{trn}}\mathbf{D}'_{\mathrm{trn}}(\mathbf{PX}))^\dagger = ((\mathbf{D}'_{\mathrm{trn}}(\mathbf{PX}))^\top (\mathbf{D}'_{\mathrm{trn}}(\mathbf{PX})))^\dagger \tag{31}$$
by Lemma 4.

By definition of Moore–Penrose pseudoinverse matrix,
$$((\mathbf{D}'_{\mathrm{trn}}(\mathbf{PX}))^\top (\mathbf{D}'_{\mathrm{trn}}(\mathbf{PX})))^\dagger = (\mathbf{D}'_{\mathrm{trn}}(\mathbf{PX}))^\top ((\mathbf{D}'_{\mathrm{trn}}(\mathbf{PX}))(\mathbf{D}'_{\mathrm{trn}}(\mathbf{PX}))^\top)^{-2} (\mathbf{D}'_{\mathrm{trn}}(\mathbf{PX}))$$
$$\tag{32}$$
where $(\mathbf{D}'_{\mathrm{trn}}(\mathbf{PX}))(\mathbf{D}'_{\mathrm{trn}}(\mathbf{PX}))^\top$ has inverse matrix by Lemma 6.

By Lemma 7,
$$\mathbf{K}^\dagger \approx \frac{1}{\epsilon}(\mathbf{D}'_{\mathrm{trn}}(\mathbf{PX}))^\top ((\mathbf{D}'_{\mathrm{trn}}(\mathbf{PX}))(\mathbf{D}'_{\mathrm{trn}}(\mathbf{PX}))^\top)^{-1} (\mathbf{D}'_{\mathrm{trn}}(\mathbf{PX})) \tag{33}$$

Therefore,
$$\mathbf{W}^* = \mathbf{K}^\dagger \mathbf{Z}^\top \mathbf{Y}_{\mathrm{trn}} \approx \frac{1}{\epsilon}(\mathbf{D}'_{\mathrm{trn}}(\mathbf{PX}))^\top ((\mathbf{D}'_{\mathrm{trn}}(\mathbf{PX}))(\mathbf{D}'_{\mathrm{trn}}(\mathbf{PX}))^\top)^{-1} (\mathbf{D}'_{\mathrm{trn}}(\mathbf{PX}))(\mathbf{PX})^\top \mathbf{D}_{\mathrm{trn}}\mathbf{Y}$$
$$\tag{34}$$

By Lemma 4,
$$\frac{1}{\epsilon}(\mathbf{D}'_{\mathrm{trn}}(\mathbf{PX}))^\top ((\mathbf{D}'_{\mathrm{trn}}(\mathbf{PX}))(\mathbf{D}'_{\mathrm{trn}}(\mathbf{PX}))^\top)^{-1} (\mathbf{D}'_{\mathrm{trn}}(\mathbf{PX}))(\mathbf{PX})^\top \mathbf{D}_{\mathrm{trn}}\mathbf{Y} \tag{35}$$
$$= \frac{1}{\epsilon}(\mathbf{D}'_{\mathrm{trn}}(\mathbf{PX}))^\top ((\mathbf{D}'_{\mathrm{trn}}(\mathbf{PX}))(\mathbf{D}'_{\mathrm{trn}}(\mathbf{PX}))^\top)^{-1} (\mathbf{D}'_{\mathrm{trn}}(\mathbf{PX}))(\mathbf{PX})^\top \mathbf{D}'^\top_{\mathrm{trn}}\mathbf{D}'_{\mathrm{trn}}\mathbf{D}_{\mathrm{trn}}\mathbf{Y} \tag{36}$$
$$= \frac{1}{\epsilon}(\mathbf{D}'_{\mathrm{trn}}(\mathbf{PX}))^\top ((\mathbf{D}'_{\mathrm{trn}}(\mathbf{PX}))(\mathbf{D}'_{\mathrm{trn}}(\mathbf{PX}))^\top)^{-1} (\mathbf{D}'_{\mathrm{trn}}(\mathbf{PX}))(\mathbf{D}'_{\mathrm{trn}}(\mathbf{PX}))'^\top \mathbf{D}'_{\mathrm{trn}}\mathbf{D}_{\mathrm{trn}}\mathbf{Y}$$
$$\tag{37}$$
$$= \frac{1}{\epsilon}(\mathbf{D}'_{\mathrm{trn}}(\mathbf{PX}))^\top \mathbf{D}'_{\mathrm{trn}}\mathbf{D}_{\mathrm{trn}}\mathbf{Y} = \frac{1}{\epsilon}(\mathbf{PX})^\top \mathbf{D}'^\top_{\mathrm{trn}}\mathbf{D}'_{\mathrm{trn}}\mathbf{D}_{\mathrm{trn}}\mathbf{Y} = \frac{1}{\epsilon}(\mathbf{PX})^\top \mathbf{D}_{\mathrm{trn}}\mathbf{Y} \tag{38}$$
Therefore,
$$\mathbf{W}^* \approx \frac{1}{\epsilon}(\mathbf{PX})^\top \mathbf{D}_{\mathrm{trn}}\mathbf{Y} \tag{39}$$

Table 7: Layer-wise formulations of five HNNs, assuming single-layer MLPs.

| Method | Each layer |
|---|---|
| HGNN | $\mathbf{X}^{(l)} = \sigma\left(\mathbf{D}_\mathrm{v}^{-\frac{1}{2}}\mathbf{H}\mathbf{D}_\mathrm{e}^{-1}\mathbf{H}^\top\mathbf{D}_\mathrm{v}^{-\frac{1}{2}}\mathbf{X}^{(l-1)}\mathbf{W}\right)$ |
| HNHN | $\mathbf{X}^{(l)} = \sigma\left(\mathbf{D}_{\mathrm{v},\alpha}^{-1}\mathbf{H}\mathbf{D}_\mathrm{e}^\alpha\sigma\left(\mathbf{D}_{\mathrm{e},\beta}^{-1}\mathbf{H}^\top\mathbf{D}_\mathrm{v}^\beta\mathbf{X}^{(l-1)}\mathbf{W}\right)\mathbf{W}\right)$ |
| UniGCNII | $\mathbf{X}^{(l)} = \sigma\left(\left((1-\alpha)\mathbf{D}_\mathrm{v}^{-1}\mathbf{H}\tilde{\mathbf{D}}_\mathrm{e}^{-1}\mathbf{H}^\top\mathbf{X}^{(l-1)} + \alpha\mathbf{X}^{(0)}\right)\tilde{\mathbf{W}}\right)$ |
| AllDeepSet | $\mathbf{X}^{(l)} = \sigma\left(\mathbf{D}_\mathrm{v}^{-1}\mathbf{H}\sigma\left(\mathbf{D}_\mathrm{e}^{-1}\mathbf{H}^\top\sigma\left(\mathbf{X}^{(l-1)}\mathbf{W}\right)\mathbf{W}\right)\mathbf{W}\right)$ |
| ED-HNN | $\mathbf{X}^{(l)} = \sigma\left(\left((1-\alpha)\sigma\left(\mathbf{D}_\mathrm{v}^{-1}\mathbf{H}\sigma\left(\mathbf{D}_\mathrm{e}^{-1}\mathbf{H}^\top\sigma\left(\mathbf{X}^{(l-1)}\mathbf{W}\right)\mathbf{W}\right)\mathbf{W}\right) + \alpha\mathbf{X}^{(0)}\right)\mathbf{W}\right)$ |

### A.3 Proof of lemma 2

We analyze the diagonal entries of the $\mathbf{A}_1 = \mathbf{D}_\mathrm{v}^{-\frac{1}{2}}\mathbf{H}\mathbf{D}_\mathrm{e}^{-1}\mathbf{H}^\top\mathbf{D}_\mathrm{v}^{-\frac{1}{2}}$. Specifically,

$$(\mathbf{A}_1)_{ii} = \frac{1}{\sqrt{d_{v_i}}}(\mathbf{H}\mathbf{D}_\mathrm{e}^{-1}\mathbf{H}^\top)_{ii}\frac{1}{\sqrt{d_{v_i}}} = \frac{1}{d_{v_i}}(\mathbf{H}\mathbf{D}_\mathrm{e}^{-1}\mathbf{H}^\top)_{ii} \tag{40}$$

Since $\mathbf{H}_{ij} = 1$ if and only if node $v_i$ belongs to hyperedge $e_j$, $(i, i)$-th entry of $\mathbf{H}\mathbf{D}_\mathrm{e}^{-1}\mathbf{H}^\top$ corresponds to the sum of edge-normalized weights over all hyperedges incident to node $v_i$. Therefore, we obtain:

$$(\mathbf{H}\mathbf{D}_\mathrm{e}^{-1}\mathbf{H}^\top)_{ii} = \sum_{e_j \in \mathcal{N}(v_i)} \frac{1}{d_{e_j}} \tag{41}$$

## B Linearization of five representative HNNs

Table 7 presents the layer-wise formulations of five hypergraph neural networks (HNNs), where each MLP block is assumed to be a single-layer perceptron. By removing nonlinear activation functions, all weight matrices in a network can be merged into a single equivalent weight matrix, since multi-layer perceptrons reduce to a linear transformation without nonlinearity.

Let $\mathbf{D}_\mathrm{v}$ and $\mathbf{D}_\mathrm{e}$ be the diagonal degree matrices of nodes and hyperedges, respectively. We define the variants of degree matrices as follows:

$$(\tilde{\mathbf{D}}_\mathrm{e})_{ii} = d_{e_i}^{-1}\sum_{v_j \in \mathcal{N}(e_i)} d_{v_j}, \quad (\mathbf{D}_{\mathrm{v},\alpha})_{ii} = \sum_{e_j \in \mathcal{N}(v_i)} d_{e_j}^\alpha, \quad (\mathbf{D}_{\mathrm{e},\beta})_{ii} = \sum_{v_j \in \mathcal{N}(e_i)} d_{v_j}^\beta.$$

We denote by $\sigma$ a nonlinear function (e.g., ReLU), and by $\tilde{\mathbf{W}} = (1-\beta)\mathbf{I}+\beta\mathbf{W}$ a convex combination of the identity and a learnable weight matrix. Since $\mathbf{W}$ is a free parameter being updated during the training, we can safely replace $\tilde{\mathbf{W}}$ with $\mathbf{W}$ under linearization.

## C Error bounds in Theorem 1

We present the relative Frobenius norm error, confirming it remains sufficiently low as $\epsilon$ decreases. In TCS, our approximation is given as follows:

$$\frac{1}{\lambda_1^2}\mathbf{M}_1 + \frac{1}{\lambda_2^2}\mathbf{M}_2 + \frac{1}{\lambda_3^2}\mathbf{M}_3 \approx \frac{1}{\epsilon}\left(\frac{1}{\lambda_1}\mathbf{M}_1 + \frac{1}{\lambda_2}\mathbf{M}_2 + \frac{1}{\lambda_3}\mathbf{M}_3\right) \tag{42}$$

where $\lambda_1 = \epsilon, \lambda_2 = (1 - 2\epsilon)k + \epsilon, \lambda_3 = k(1 - 2\epsilon + \epsilon c) + \epsilon, \mathbf{M}_1 = \mathbf{I}_{kc} - \frac{1}{k}(\mathbf{I}_c \otimes \mathbf{J}_k), \mathbf{M}_2 = \frac{1}{k}(\mathbf{I}_c \otimes \mathbf{J}_k) - \frac{1}{kc}\mathbf{J}_{kc}, \mathbf{M}_3 = \frac{1}{kc}\mathbf{J}_{kc}$. $\mathbf{J}$ denotes an all-one matrix and $\otimes$ is the Kronecker product.

To evaluate the approximation quality, we compute the relative Frobenius norm error as follows:

$$\frac{\left|\frac{1}{\lambda_1^2}\mathbf{M}_1 + \frac{1}{\lambda_2^2}\mathbf{M}_2 + \frac{1}{\lambda_3^2}\mathbf{M}_3 - \frac{1}{\epsilon}\left(\frac{1}{\lambda_1}\mathbf{M}_1 + \frac{1}{\lambda_2}\mathbf{M}_2 + \frac{1}{\lambda_3}\mathbf{M}_3\right)\right|_F}{\left|\frac{1}{\lambda_1^2}\mathbf{M}_1 + \frac{1}{\lambda_2^2}\mathbf{M}_2 + \frac{1}{\lambda_3^2}\mathbf{M}_3\right|_F} \tag{43}$$

Table 8: Relative error for different values of $\epsilon$.

| $\epsilon$ | Relative Error |
|---|---|
| 0.1 | 1.14% |
| 0.01 | 0.10% |
| 0.001 | 0.01% |

For a representative case with $k = 5$ and $c = 10$, the relative errors are summarized in Table 8.

These results indicate that the approximation becomes increasingly accurate as $\epsilon$ decreases. This supports the validity of our approach, particularly in the regime where $\epsilon$ is sufficiently small.

## D   Row-normalized adjacency matrices

Eq. (44) and Eq. (45) present the row-normalized adjacency matrices $\hat{\mathbf{A}}_1$ and $\hat{\mathbf{A}}_2$. The corresponding residual self-information terms, $\mathrm{RSI}(\hat{\mathbf{A}}_1)$ and $\mathrm{RSI}(\hat{\mathbf{A}}_2)$, are shown to coincide with those obtained under symmetric normalization. This equivalence is formally established in Lemma 8 and Lemma 9.

$$\hat{\mathbf{A}}_1 = \mathbf{D}_{\mathrm{v}}^{-1}\mathbf{H}(\mathbf{D}_{\mathrm{e}} - \mathbf{I})^{-1}\mathbf{H}^\top \tag{44}$$

$$\hat{\mathbf{A}}_2 = \mathbf{A}_1^* \left((\mathbf{D}_{\mathrm{v}} - \mathbf{I})^{-1}\mathbf{D}_{\mathrm{v}}\right) \mathbf{A}_1^*. \tag{45}$$

**Lemma 8.** *Given $\hat{\mathbf{A}}_1$ in Eq. (44), $\mathrm{RSI}(\hat{\mathbf{A}}_1)$ is given by*

$$(\mathrm{RSI}(\hat{\mathbf{A}}_1))_{ii} = d_{v_i}^{-1} \left(\sum_{e_j \in \mathcal{N}(v_i)}(d_{e_j} - 1)^{-1}\right), \tag{46}$$

*where $d_x$ denotes the degree of node $x$ or the number of nodes in hyperedge $x$, based on the type of $x$, and $\mathcal{N}(v_i)$ denotes the set of hyperedges incident to node $v_i$.*

*Proof.* We analyze the diagonal entries of the $\mathbf{A}_1 = \mathbf{D}_{\mathrm{v}}^{-1}\mathbf{H}\mathbf{D}_{\mathrm{e}}^{-1}\mathbf{H}^\top$. Specifically,

$$(\mathbf{A}_1)_{ii} = \frac{1}{d_{v_i}}(\mathbf{H}\mathbf{D}_{\mathrm{e}}^{-1}\mathbf{H}^\top)_{ii} \tag{47}$$

Since $\mathbf{H}_{ij} = 1$ if and only if node $v_i$ belongs to hyperedge $e_j$, $(i, i)$-th entry of $\mathbf{H}\mathbf{D}_{\mathrm{e}}^{-1}\mathbf{H}^\top$ corresponds to the sum of edge-normalized weights over all hyperedges incident to node $v_i$. Therefore, we obtain:

$$(\mathbf{H}\mathbf{D}_{\mathrm{e}}^{-1}\mathbf{H}^\top)_{ii} = \sum_{e_j \in \mathcal{N}(v_i)} \frac{1}{d_{e_j}} \tag{48}$$

$\square$

**Lemma 9.** *Given $\mathbf{A}_1^* = \hat{\mathbf{A}}_1 - \mathrm{RSI}(\hat{\mathbf{A}}_1)$ and $\hat{\mathbf{A}}_2$ in Eq. (45), $\mathrm{RSI}(\hat{\mathbf{A}}_2)$ is given by*

$$(\mathrm{RSI}(\hat{\mathbf{A}}_2))_{ii} = d_{v_i}^{-1} \left(\sum_{e_j \in \mathcal{N}(v_i)}(d_{e_j} - 1)^{-2} \left(\sum_{v_k \in \mathcal{N}(e_j) \setminus \{v_i\}}(d_{v_k} - 1)^{-1}\right)\right), \tag{49}$$

*where $d_x$ denotes the degree of node $x$ or the number of nodes in hyperedge $x$, based on the type of $x$, $\mathcal{N}(v_i)$ denotes the set of hyperedges incident to node $v_i$, and $\mathcal{N}(e_j)$ denotes the set of nodes incident to hyperedge $e_j$.*

*Proof.* The proof follows by applying the same reasoning as in Lemma 8. $\square$

## E   Possible approximations for deeper propagation

Some datasets may require models with higher-hop propagations for better expressivity. Without approximation, the exact computation of RSI requires high computational cost. One possible approximation is to estimate the probability that a random walker returns to the starting node after steps, where is the number of hops that we aim to model. A simple pseudocode for this strategy is provided in Algorithm 1.

---

**Algorithm 1** Approximation via random walks

---

**Require:** Node $i$, walk length $l$
**Ensure:** Return probability of node $i$
 1: $count \leftarrow 0$
 2: **for** each trial **do**
 3:  $current \leftarrow i$
 4:  **for** $t = 1$ **to** $l$ **do**
 5:    Sample a hyperedge $e$ incident to $current$ uniformly at random
 6:    Sample a node $j$ connected to $e$ uniformly at random
 7:    $current \leftarrow j$
 8:  **end for**
 9:  **if** $current = i$ **then**
10:    $count \leftarrow count + 1$
11:  **end if**
12: **end for**
13: **return** $count/(\text{number of trials})$

---

Another possible approximation is to use Hutchinson's Estimator, which provides an unbiased stochastic estimate of the diagonal entries. The equation is given by:

$$\text{diag}(\mathbf{A}) \approx \frac{1}{m} \sum_{k=1}^{m} \mathbf{z}^{(k)} \odot \left( \mathbf{A}\mathbf{z}^{(k)} \right) \tag{50}$$

where $m$ is the number of random probe vectors, each $\mathbf{z}^{(k)} \in \mathbb{R}^n$ is a random vector with entries independently sampled from the Rademacher distribution (i.e., each entry is $+1$ or $-1$ with equal probability), and $\odot$ denotes the element-wise (Hadamard) product. The expectation satisfies $\text{diag}(\mathbf{A}) = \mathbb{E}[\mathbf{z} \odot (\mathbf{A}\mathbf{z})]$.

Hutchinson's Estimator does not require explicit storage of the entire matrix $\mathbf{A}$; instead, it relies solely on matrix-vector multiplications with randomized vectors $\mathbf{z}$. This characteristic renders Hutchinson's Estimator particularly suitable and computationally efficient for hypergraph structures. We consider these approaches to be promising directions for extending ZEN to scenarios where deeper propagation may provide additional benefits.

# F  Additional experiments

In this section, we present additional experimental results omitted from the main paper due to space limitations.

## F.1  Evaluation with increasing shots

ZEN is a parameter-free model, which gives it a particular advantage in settings where labeled data is scarce and training is challenging. While ZEN was orginally designed for few-shot setting, our results in Table 9 and Table 10 show it performs strongly with more training samples as well. ZEN achieves top average ranks in both 10-shot and 20-shot settings, suggesting it scales well beyond few-shot scenarios.

## F.2  Evaluation against additional baselines

We have additionally included eight baselines; three representative linear GNNs (SGC [26], APPNP [7], and SSGC [35]), three linearized hypergraph models based on UniGCNII [10], AllDeepSets [3], and ED-HNN [23], and two semi-supervised hypergraph models (LEGCN [30] and HyperND [22]). LEGCN utilizes a line expansion approach to adapt hypergraphs to conventional GNN architectures, while HyperND introduces a diffusion-based mechanism for improved label propagation in hypergraphs. For GNNs, we applied clique expansion to convert the hypergraph into a pairwise graph. As shown in the Table 11, ZEN demonstrates consistently strong performance, outperforming all linear GNN and linearized hypergraph baselines on the majority of datasets.

Table 9: Classification accuracy (%) for 10-shot node classification on real-world hypergraphs. We report the mean and standard deviation over 10 runs. Boldfaced letters indicate the best accuracy, and underlined letters indicate the second. ZEN achieves the highest average rank.

| Methods | Cora | Citeseer | Pubmed | Cora_CA | 20News | MN40 | Congress | Wallmart | Senate | House | Avg. Rank |
|---|---|---|---|---|---|---|---|---|---|---|---|
| HGNN | $59.13_{\pm5.8}$ | $49.28_{\pm3.1}$ | $60.53_{\pm7.0}$ | $64.94_{\pm4.4}$ | $75.54_{\pm1.2}$ | $94.90_{\pm0.2}$ | $87.18_{\pm0.9}$ | $46.85_{\pm1.2}$ | $58.01_{\pm2.5}$ | $64.30_{\pm4.4}$ | 5.4 |
| HNHN | $42.47_{\pm5.0}$ | $42.82_{\pm2.8}$ | $58.92_{\pm3.3}$ | $46.69_{\pm3.5}$ | $51.23_{\pm4.9}$ | $93.96_{\pm0.6}$ | $47.77_{\pm1.8}$ | $17.13_{\pm2.2}$ | $69.74_{\pm6.1}$ | $70.32_{\pm7.6}$ | 8.0 |
| HCHA | $58.96_{\pm5.4}$ | $49.65_{\pm2.9}$ | $60.02_{\pm5.6}$ | $64.28_{\pm5.6}$ | $\mathbf{75.59_{\pm1.3}}$ | $94.89_{\pm0.2}$ | $87.23_{\pm0.9}$ | $47.11_{\pm1.7}$ | $57.89_{\pm2.8}$ | $63.86_{\pm4.4}$ | 5.7 |
| UniGCN | $\underline{60.87_{\pm5.8}}$ | $51.13_{\pm3.0}$ | $61.20_{\pm6.9}$ | $\underline{66.87_{\pm3.0}}$ | $73.69_{\pm1.7}$ | $96.24_{\pm0.2}$ | $\mathbf{91.35_{\pm0.7}}$ | $45.38_{\pm1.4}$ | $62.57_{\pm2.5}$ | $70.27_{\pm3.0}$ | $\underline{3.7}$ |
| UniGCNII | $56.81_{\pm5.1}$ | $49.18_{\pm2.9}$ | $61.12_{\pm6.4}$ | $63.17_{\pm3.8}$ | $70.91_{\pm2.1}$ | $97.03_{\pm0.2}$ | $87.72_{\pm1.9}$ | $28.70_{\pm2.5}$ | $\mathbf{75.14_{\pm4.3}}$ | $73.65_{\pm2.9}$ | 5.1 |
| AllDeepSets | $57.31_{\pm4.4}$ | $50.63_{\pm2.6}$ | $61.39_{\pm5.1}$ | $62.78_{\pm4.1}$ | $62.15_{\pm2.3}$ | $95.25_{\pm0.3}$ | $72.63_{\pm5.8}$ | $35.54_{\pm2.7}$ | $69.53_{\pm6.7}$ | $69.96_{\pm3.1}$ | 6.3 |
| AllSetTransformer | $57.50_{\pm5.0}$ | $\underline{54.07_{\pm2.4}}$ | $\underline{63.41_{\pm6.6}}$ | $\mathbf{67.63_{\pm4.2}}$ | $72.67_{\pm1.4}$ | $95.87_{\pm0.1}$ | $83.09_{\pm2.2}$ | $45.39_{\pm3.7}$ | $73.13_{\pm5.5}$ | $70.99_{\pm4.5}$ | 3.9 |
| ED-HNN | $58.84_{\pm4.3}$ | $51.12_{\pm2.8}$ | $60.59_{\pm7.0}$ | $64.73_{\pm3.7}$ | $71.13_{\pm2.2}$ | $96.30_{\pm0.1}$ | $\underline{90.73_{\pm1.8}}$ | $\mathbf{47.17_{\pm2.3}}$ | $66.44_{\pm9.3}$ | $62.49_{\pm7.9}$ | 4.7 |
| ZEN (proposed) | $\mathbf{61.44_{\pm4.2}}$ | $\mathbf{59.17_{\pm2.3}}$ | $\mathbf{68.76_{\pm3.4}}$ | $65.60_{\pm3.9}$ | $73.04_{\pm2.4}$ | $\mathbf{97.92_{\pm0.1}}$ | $86.54_{\pm2.6}$ | $\underline{47.12_{\pm3.2}}$ | $\underline{74.72_{\pm1.9}}$ | $\mathbf{73.90_{\pm6.4}}$ | 2.2 |

Table 10: Classification accuracy (%) for 20-shot node classification on real-world hypergraphs. We report the mean and standard deviation over 10 runs. Boldfaced letters indicate the best accuracy, and underlined letters indicate the second. ZEN achieves the highest average rank.

| Methods | Cora | Citeseer | Pubmed | Cora_CA | 20News | MN40 | Congress | Wallmart | Senate | House | Avg. Rank |
|---|---|---|---|---|---|---|---|---|---|---|---|
| HGNN | $62.46_{\pm3.1}$ | $56.57_{\pm1.9}$ | $69.65_{\pm3.9}$ | $70.33_{\pm1.9}$ | $\underline{76.66_{\pm1.1}}$ | $95.07_{\pm0.2}$ | $87.80_{\pm0.5}$ | $49.46_{\pm2.4}$ | $60.73_{\pm3.2}$ | $64.83_{\pm2.2}$ | 6.1 |
| HNHN | $48.28_{\pm3.9}$ | $50.08_{\pm2.2}$ | $62.60_{\pm1.6}$ | $52.88_{\pm2.9}$ | $52.69_{\pm6.3}$ | $95.20_{\pm0.5}$ | $49.92_{\pm3.0}$ | $21.82_{\pm1.8}$ | $73.51_{\pm3.9}$ | $72.00_{\pm4.1}$ | 8.0 |
| HCHA | $64.19_{\pm3.9}$ | $56.81_{\pm2.1}$ | $69.44_{\pm3.3}$ | $70.04_{\pm1.9}$ | $\mathbf{76.68_{\pm1.0}}$ | $95.09_{\pm0.1}$ | $87.91_{\pm0.4}$ | $49.44_{\pm2.2}$ | $61.07_{\pm3.3}$ | $64.91_{\pm2.3}$ | 5.7 |
| UniGCN | $\underline{65.44_{\pm3.3}}$ | $58.47_{\pm1.2}$ | $\underline{70.84_{\pm2.6}}$ | $\underline{71.26_{\pm1.8}}$ | $72.59_{\pm2.1}$ | $96.63_{\pm0.1}$ | $\mathbf{91.62_{\pm0.3}}$ | $49.00_{\pm2.1}$ | $62.04_{\pm2.3}$ | $72.69_{\pm0.9}$ | 3.9 |
| UniGCNII | $64.29_{\pm2.9}$ | $56.97_{\pm1.2}$ | $68.92_{\pm3.4}$ | $68.55_{\pm2.5}$ | $74.05_{\pm1.5}$ | $\underline{97.56_{\pm0.1}}$ | $88.31_{\pm1.9}$ | $32.04_{\pm1.5}$ | $\mathbf{77.59_{\pm2.0}}$ | $\underline{76.05_{\pm0.9}}$ | 4.4 |
| AllDeepSets | $62.37_{\pm3.7}$ | $57.78_{\pm1.8}$ | $67.10_{\pm2.2}$ | $65.93_{\pm1.3}$ | $67.60_{\pm2.5}$ | $96.69_{\pm0.3}$ | $77.65_{\pm5.8}$ | $43.97_{\pm2.4}$ | $73.05_{\pm2.2}$ | $73.29_{\pm2.5}$ | 6.4 |
| AllSetTransformer | $63.65_{\pm1.6}$ | $\underline{59.49_{\pm2.1}}$ | $69.70_{\pm2.3}$ | $69.82_{\pm2.2}$ | $74.22_{\pm1.2}$ | $96.02_{\pm0.2}$ | $89.20_{\pm0.5}$ | $49.29_{\pm1.8}$ | $\underline{76.14_{\pm3.8}}$ | $75.58_{\pm2.7}$ | $\underline{3.8}$ |
| ED-HNN | $64.24_{\pm3.0}$ | $57.27_{\pm1.6}$ | $69.56_{\pm2.3}$ | $69.29_{\pm1.6}$ | $70.92_{\pm1.9}$ | $96.69_{\pm0.2}$ | $\underline{90.11_{\pm2.1}}$ | $50.61_{\pm3.3}$ | $72.30_{\pm6.7}$ | $71.27_{\pm3.7}$ | 4.8 |
| ZEN (proposed) | $\mathbf{67.87_{\pm1.4}}$ | $\mathbf{64.00_{\pm1.4}}$ | $\mathbf{71.32_{\pm1.6}}$ | $\mathbf{71.99_{\pm2.0}}$ | $73.74_{\pm1.2}$ | $\mathbf{98.21_{\pm0.1}}$ | $88.04_{\pm2.9}$ | $\mathbf{51.30_{\pm1.1}}$ | $74.10_{\pm5.2}$ | $\mathbf{76.70_{\pm0.4}}$ | 2.0 |

Table 11: Comparison of classification accuracy (%) with eight additional baselines for 5-shot node classification on real-world hypergraphs. We report the mean and standard deviation over 10 runs. Boldfaced letters indicate the best accuracy, and underlined letters indicate the second. ZEN achieves the highest average rank.

| Methods | Cora | Citeseer | Pubmed | Cora_CA | 20News | MN40 | Congress | Wallmart | Senate | House | Avg. Rank |
|---|---|---|---|---|---|---|---|---|---|---|---|
| SGC | $44.09_{\pm9.8}$ | $40.42_{\pm5.3}$ | $57.19_{\pm5.5}$ | $50.79_{\pm5.9}$ | $58.06_{\pm5.1}$ | $91.58_{\pm0.7}$ | $72.48_{\pm3.1}$ | $24.11_{\pm3.2}$ | $51.26_{\pm2.7}$ | $52.27_{\pm1.4}$ | 5.9 |
| APPNP | $\underline{45.77_{\pm9.5}}$ | $39.55_{\pm4.7}$ | $55.38_{\pm7.1}$ | $50.04_{\pm6.7}$ | $59.90_{\pm5.8}$ | $92.75_{\pm0.3}$ | $69.09_{\pm6.8}$ | $24.70_{\pm3.3}$ | $71.24_{\pm5.4}$ | $70.00_{\pm8.4}$ | 5.8 |
| SSGC | $42.60_{\pm10.9}$ | $40.69_{\pm4.7}$ | $57.07_{\pm4.9}$ | $\underline{52.55_{\pm6.7}}$ | $59.91_{\pm5.2}$ | $93.75_{\pm0.2}$ | $81.10_{\pm2.4}$ | $26.56_{\pm3.4}$ | $73.00_{\pm0.7}$ | $71.74_{\pm3.2}$ | $\underline{3.3}$ |
| lin UniGCNII | $39.44_{\pm7.4}$ | $40.00_{\pm4.0}$ | $56.85_{\pm4.9}$ | $49.75_{\pm6.1}$ | $59.45_{\pm5.2}$ | $96.58_{\pm0.4}$ | $73.35_{\pm3.1}$ | $17.91_{\pm1.7}$ | $71.22_{\pm10.8}$ | $68.23_{\pm10.6}$ | 5.8 |
| lin AllDeepSets | $40.49_{\pm8.2}$ | $\underline{40.76_{\pm4.5}}$ | $56.13_{\pm6.0}$ | $51.04_{\pm5.8}$ | $\underline{66.62_{\pm5.0}}$ | $96.62_{\pm0.3}$ | $\mathbf{90.39_{\pm1.5}}$ | $26.45_{\pm3.0}$ | $60.95_{\pm4.7}$ | $65.13_{\pm13.1}$ | 4.3 |
| lin ED-HNN | $41.97_{\pm5.6}$ | $39.64_{\pm4.2}$ | $55.82_{\pm5.4}$ | $47.37_{\pm5.8}$ | $61.47_{\pm6.0}$ | $\underline{97.28_{\pm0.4}}$ | $78.35_{\pm2.7}$ | $18.84_{\pm1.6}$ | $71.10_{\pm9.9}$ | $70.20_{\pm9.3}$ | 5.2 |
| LEGCN | $37.59_{\pm5.2}$ | $37.25_{\pm3.8}$ | $\underline{58.11_{\pm4.0}}$ | $37.59_{\pm5.2}$ | $49.41_{\pm4.3}$ | $93.27_{\pm0.7}$ | $72.14_{\pm3.4}$ | O.O.M | $71.25_{\pm8.5}$ | $\underline{72.85_{\pm7.0}}$ | 6.5 |
| HyperND | $39.09_{\pm6.2}$ | $35.26_{\pm4.8}$ | $56.54_{\pm4.5}$ | $41.74_{\pm4.6}$ | $54.70_{\pm3.8}$ | $91.40_{\pm0.5}$ | $73.57_{\pm4.9}$ | $13.55_{\pm1.7}$ | $\mathbf{73.88_{\pm7.5}}$ | $72.84_{\pm7.1}$ | 6.5 |
| ZEN (proposed) | $\mathbf{51.85_{\pm10.1}}$ | $\mathbf{49.08_{\pm4.8}}$ | $\mathbf{62.62_{\pm3.9}}$ | $\mathbf{60.04_{\pm6.2}}$ | $\mathbf{68.57_{\pm4.8}}$ | $\mathbf{97.63_{\pm0.3}}$ | $\underline{86.96_{\pm4.8}}$ | $\mathbf{43.88_{\pm3.1}}$ | $70.40_{\pm10.0}$ | $\mathbf{73.22_{\pm6.3}}$ | 1.7 |

### F.3 Evaluation under standard *n*-way *k*-shot setting

To examine the generality of our approach, we further evaluate it under the standard *n*-way *k*-shot setting. To the best of our knowledge, this setup has not been explicitly explored for general hypergraphs. Accordingly, we designed a new evaluation protocol by drawing inspiration from the experimental setup of Meta-GNN [34] and adopting hyperparameter configurations aligned with those used in ED-HNN [23]. The results, summarized in Table 12, demonstrate the strong performance of ZEN in this setting, underscoring its effectiveness for few-shot learning on hypergraphs. Notably, ZEN achieves competitive results even when applied solely to test episodes, revealing a promising and underexplored direction for future research.

## G Running times

The actual running times are reported in Table 13, with all values measured in seconds. ZEN runs in less than a second across all datasets, achieving as fast as 0.003s on the Senate dataset.

Table 12: Classification accuracy (%) under the standard *n*-way *k*-shot setting. Results are averaged over 10 runs. ZEN achieves superior performance across all configurations.

| Methods | Cora (2-way 3-shot) | Cora (2-way 1-shot) | Citeseer (2-way 3-shot) | Citeseer (2-way 1-shot) |
|---|---|---|---|---|
| HGNN | $64.4_{\pm 0.1}$ | $55.9_{\pm 0.1}$ | $60.1_{\pm 0.2}$ | $54.9_{\pm 0.2}$ |
| UniGCNII | $68.0_{\pm 0.1}$ | $58.9_{\pm 0.2}$ | $65.2_{\pm 0.1}$ | $56.8_{\pm 1.6}$ |
| AllDeepSets | $52.3_{\pm 0.2}$ | $48.7_{\pm 0.2}$ | $51.1_{\pm 0.2}$ | $50.2_{\pm 0.2}$ |
| **ZEN (proposed)** | $\mathbf{73.3}_{\pm 0.1}$ | $\mathbf{62.6}_{\pm 0.1}$ | $\mathbf{71.5}_{\pm 0.1}$ | $\mathbf{62.3}_{\pm 0.1}$ |

Table 13: The actual running time of ZEN and the baseline models, including both training and inference. Each time is measured in seconds. ZEN exhibits a significant speed advantage over all baselines.

| Methods | Cora | Citeseer | Pubmed | Cora_CA | 20News | MN40 | Congress | Walmart | Senate | House |
|---|---|---|---|---|---|---|---|---|---|---|
| HGNN | 2.301 | 2.800 | 2.423 | 2.506 | 2.499 | 3.958 | 4.950 | 20.277 | 2.540 | 2.543 |
| HNHN | 2.023 | 1.996 | 2.877 | 2.024 | 3.676 | 2.720 | 3.925 | 14.435 | 2.278 | 2.267 |
| HCHA | 3.290 | 3.485 | 4.448 | 2.850 | 2.894 | 4.072 | 6.675 | 8.558 | 3.297 | 4.584 |
| UniGCN | 4.553 | 1.932 | 6.693 | 2.511 | 6.825 | 4.315 | 8.497 | 22.121 | 2.342 | 1.920 |
| UniGCNII | 4.193 | 3.812 | 2.069 | 4.826 | 4.025 | 3.831 | 7.472 | 24.452 | 2.277 | 2.419 |
| AllDeepSets | 15.456 | 18.331 | 8.889 | 16.108 | 8.420 | 9.378 | 25.499 | 60.619 | 13.232 | 11.461 |
| AllSetTransformer | 2.597 | 3.167 | 8.865 | 3.462 | 14.273 | 4.924 | 6.114 | 51.702 | 3.260 | 3.435 |
| ED-HNN | 4.267 | 4.458 | 4.407 | 6.588 | 11.072 | 5.664 | 39.791 | 31.137 | 2.336 | 2.487 |
| **ZEN (proposed)** | **0.266** | **0.736** | **0.807** | **0.280** | **0.237** | **0.197** | **0.093** | **0.672** | **0.003** | **0.007** |

