# OpenReview forum: "Parameter-Free Hypergraph Neural Network for Few-Shot Node Classification"
_NeurIPS.cc/2025/Conference — NeurIPS 2025 poster_

### Official Review · Reviewer_UPmr · 2025-06-24

**Clarity:** 3
**Significance:** 3
**Originality:** 3
**Rating:** 5
**Confidence:** 3

**Summary:**

The paper proposes a novel parameter-free hypergraph neural network for few-shot node classification. Tractable Closed-form Solution (TCS) eliminates iterative training, and Redundancy-Aware Propagation (RAP) explicitly removes self-information redundancy. Comprehensive tests show improved accuracy and efficiency compared to previous methods.

**Questions:**

* Can the authors provide error bounds for closed-form approximation in Theorem 1?
* Can the authors clarify the unit used in Table 4?

**Ethical Concerns:**

["NO or VERY MINOR ethics concerns only"]

**Final Justification:**

The rebuttal answers all my questions, so I keep my score.

**Limitations:**

yes

**Paper Formatting Concerns:**

Formatting instructions are followed.

**Quality:**

3

**Strengths And Weaknesses:**

**Strengths**
* The method shows improved accuracy and efficiency compared to previous methods.
* The paper is technically solid with rigorous theoretical analysis.
* The paper is well-written and easy to read.

**Weakness**
* There is no error bounds for closed-form approximation in Theorem 1.
* Table 4 presents values without specifying units. If the time for the proposed method is used as the unit of time, please state this clearly and maybe provide the original data in the appendix.
* Typos exist. For example, $W_*$ should be $W^*$ in Proof of lemma 1.

---

> ### Author Rebuttal · Authors · 2025-07-31
>
> We greatly appreciate your high recognition of our work, and are eager to share our thoughts with you.
>
> **Response to Question 1:**
> > **Question 1:** Can the authors provide error bounds for closed-form approximation in Theorem 1?
>
> Yes, we believe more discussion on the error bounds can make our theoretical claim stronger. We present a summary of our analysis as follows, which will be included in the final version of our paper as well.
>
> In TCS, our approximation is given as follows:
> $$
> \frac{1}{\lambda_1^2}M_1+\frac{1}{\lambda_2^2}M_2+\frac{1}{\lambda_3^2}M_3 \approx \frac{1}{\epsilon}\left(\frac{1}{\lambda_1}M_1+\frac{1}{\lambda_2}M_2+\frac{1}{\lambda_3}M_3\right),
> $$
> where
> - $\lambda_1 = \epsilon$
> - $\lambda_2 = (1 - 2\epsilon)k + \epsilon$
> - $\lambda_3 = k(1 - 2\epsilon + \epsilon c) + \epsilon$
>
> and the matrices form an orthogonal decomposition:
> - $M_1 = I_{kc} - \frac{1}{k}(I_c \otimes J_k)$
> - $M_2 = \frac{1}{k}(I_c \otimes J_k) - \frac{1}{kc}J_{kc}$
> - $M_3 = \frac{1}{kc} J_{kc}$
>
> Here, $J$ denotes an all-one matrix and $\otimes$ is the Kronecker product.
>
> To evaluate the approximation quality, we compute the relative Frobenius norm error:
> $$
> \frac{\left\|\frac{1}{\lambda_1^2}M_1+\frac{1}{\lambda_2^2}M_2+\frac{1}{\lambda_3^2}M_3 - \frac{1}{\epsilon}\left(\frac{1}{\lambda_1}M_1+\frac{1}{\lambda_2}M_2+\frac{1}{\lambda_3}M_3\right)\right\|_F}{\left\|\frac{1}{\lambda_1^2}M_1+\frac{1}{\lambda_2^2}M_2+\frac{1}{\lambda_3^2}M_3\right\|_F}
> $$
>
> For a representative case with $k = 5$, $c = 10$, the relative errors are as follows:
>
> | $\epsilon$ | Relative Error |
> |------------|----------------|
> | 0.1        | 1.14%          |
> | 0.01       | 0.10%          |
> | 0.001      | 0.01%          |
>
> These results indicate that the approximation becomes increasingly accurate as $\epsilon$ decreases. This supports the validity of our approach, particularly in the regime where $\epsilon$ is sufficiently small.
>
> **Response to Question 2:**
> > **Question 2:** Can the authors clarify the unit used in Table 4?
>
> Thank you for your suggestion. As shown in Table 4, the values represent the running times relative to ZEN, and thus no explicit unit is indicated. The actual running times are reported in the table below, with all values measured in seconds (s). ZEN runs in less than a second across all datasets, achieving as fast as 0.003s on the Senate dataset.
>
> |Method|Cora|Citeseer|Pubmed|Cora_ca|20News|MN40|Congress|Walmart|Senate|House|
> |:-|-|-|-|-|-|-|-|-|-|-|
> |HGNN|2.301|2.800|2.423|2.506|2.499|3.958|4.950|20.277|2.540|2.543|
> |HNHN|2.023|1.996|2.877|2.024|3.676|2.720|3.925|14.435|2.278|2.267|
> |HCHA|3.290|3.485|4.448|2.850|2.894|4.072|6.675|8.558|3.297|4.584|
> |UniGCN|4.553|1.932|6.693|2.511|6.825|4.315|8.497|22.121|2.342|1.920|
> |UniGCNII|4.193|3.812|2.069|4.826|4.025|3.831|7.472|24.452|2.277|2.419|
> |AllDeepSets|15.456|18.331|8.889|16.108|8.420|9.378|25.499|60.619|13.232|11.461|
> |AllSetTransformer|2.597|3.167|8.865|3.462|14.273|4.924|6.114|51.702|3.260|3.435|
> |ED-HNN|4.267|4.458|4.407|6.588|11.072|5.664|39.791|31.137|2.336|2.487|
> |ZEN (**ours**)|**0.266**|**0.736**|**0.807**|**0.280**|**0.237**|**0.197**|**0.093**|**0.672**|**0.003**|**0.007**|

---

> > ### Comment · Reviewer_UPmr · 2025-08-05
> >
> > Thanks for answering all my questions. I'll keep my score.

---

> > > ### Author Response · Authors · 2025-08-06
> > >
> > > Thank you again for your insightful review. We sincerely appreciate your time and consideration.

---

### Official Review · Reviewer_SfcC · 2025-06-27

**Clarity:** 3
**Significance:** 3
**Originality:** 4
**Rating:** 5
**Confidence:** 2

**Summary:**

This paper proposes ZEN, which is a parameter-free hypergraph neural network. They theoretically derive their parameter-free solution. The experimental study shows the effectiveness and efficiency of ZEN compared with 8 baselines in ten datasets.

**Questions:**

Q1. Why do the authors focus on few-shot classifications? Is it possible to use ZEN for non-few-shot settings?

Q2. Could you add the actual running time in ZEN?

**Ethical Concerns:**

["NO or VERY MINOR ethics concerns only"]

**Final Justification:**

This paper provides a theoretical analysis of parameter-free models and empirical experiments that showed their performance. Their rebuttals clarified the issues of presentations. I believe that this paper is ready to be accepted in NeurIPS.

**Limitations:**

Yes, but very short.

**Paper Formatting Concerns:**

nan

**Quality:**

3

**Strengths And Weaknesses:**

Strong points.

S1. It is surprising that the parameter-free method ZEN outperforms hypergraph neural networks with learning parameters.

S2. Their theoretical analysis supports their structural design of ZEN.

S3. The experimental study is sufficient, validating efficiency and effectiveness in ten datasets, analyzing the effectiveness of components in ZEN as an ablation study, and testing interpretability.

Weak points.

W1. It is unclear why the authors focus only on few-shot node classification. ZEN seems distinct from the few-shot node classification. And, it is unclear what the performance is if the size of the training data varies.

W2. They do not show the actual running time in ZEN.

---

> ### Author Rebuttal · Authors · 2025-07-31
>
> Thank you for your recognition of our work and providing the valuable suggestions and constructive comments. Your thoughtful feedback is deeply appreciated and will greatly assist us in refining and strengthening the manuscript.
>
> **Response to Question 1:**
> > **Question 1:** Why do the authors focus on few-shot classifications? Is it possible to use ZEN for non-few-shot settings?
>
> ZEN is a parameter-free model, which gives it a particular advantage in settings where labeled data is scarce and training is challenging. While ZEN was orginally designed for few-shot setting, our results in the tables below show it performs strongly with more training samples as well. ZEN achieves top average ranks in both 10-shot and 20-shot settings, suggesting it scales well beyond few-shot scenarios.
>
> ### 10-shot classification (10 training nodes per class)
>
> |Method|Cora|Citeseer|Pubmed|Cora_ca|20News|MN40|Congress|Walmart|Senate|House|Avg. Rank|
> |:-|-|-|-|-|-|-|-|-|-|-|:-:|
> |HGNN|59.1±5.8|49.3±3.1|60.5±7.0|64.9±4.4|75.5±1.2|94.9±0.2|87.2±0.9|46.9±1.2|58.0±2.5|64.3±4.4|6.1|
> |HNHN|42.5±5.0|42.8±2.8|58.9±3.3|46.7±3.5|51.2±4.9|94.0±0.6|47.8±1.8|17.1±2.2|69.7±6.1|70.3±7.6|9.9|
> |HCHA|59.0±5.4|49.7±2.9|60.0±5.6|64.3±5.6|**75.6±1.3**|94.9±0.2|87.2±0.9|47.1±1.7|57.9±2.8|63.9±4.4|5.8|
> |UniGCN|60.9±5.8|51.1±3.0|61.2±6.9|66.9±3.0|73.7±1.7|96.2±0.2|**91.4±0.7**|45.4±1.4|62.6±2.5|70.3±3.0|4.1|
> |UniGCNII|56.8±5.1|49.2±2.9|61.1±6.4|63.2±3.8|70.9±2.1|97.0±0.2|87.7±1.9|28.7±2.5|**75.1±4.3**|73.7±2.9|4.8|
> |AllDeepSets|57.3±4.4|50.6±2.6|61.4±5.1|62.8±4.1|62.2±2.3|95.3±0.3|72.6±5.8|35.5±2.7|69.5±6.7|70.0±3.1|6.9|
> |AllSetTransformer|57.5±5.0|54.1±2.4|63.4±6.6|**67.6±4.2**|72.7±1.4|95.9±0.1|83.1±2.2|45.4±3.7|73.1±5.5|71.0±4.5|3.9|
> |ED-HNN|58.8±4.3|51.1±2.8|60.6±7.0|64.7±3.7|71.1±2.2|96.3±0.2|90.7±1.8|**47.2±2.3**|66.4±9.3|62.5±7.9|4.9|
> |ZEN (**ours**)|**61.4±4.2**|**59.2±2.3**|**68.8±3.4**|65.6±3.9|73.0±2.4|**97.9±0.1**|86.5±2.6|47.1±3.2|74.7±1.9|**73.9±6.4**|**1.7**|
>
> ### 20-shot classification (20 training nodes per class)
>
> |Method|Cora|Citeseer|Pubmed|Cora_ca|20News|MN40|Congress|Walmart|Senate|House|Avg. Rank|
> |:-|-|-|-|-|-|-|-|-|-|-|:-:|
> |HGNN|62.5±3.1|56.6±1.9|69.7±3.9|70.3±1.9|**76.7±1.1**|95.1±0.2|87.8±0.5|49.5±2.4|60.7±3.2|64.8±2.2|5.4|
> |HNHN|48.3±3.9|50.1±2.3|62.6±1.6|52.9±2.9|52.7±6.3|95.2±0.5|49.9±3.0|21.8±1.8|73.5±3.9|72.0±4.1|9.7|
> |HCHA|64.2±3.9|56.8±2.1|69.4±3.3|70.0±1.9|**76.7±1.0**|95.1±0.1|87.9±0.4|49.4±2.2|61.1±3.3|64.9±2.3|4.9|
> |UniGCN|65.4±3.3|58.5±1.2|70.8±2.6|71.3±1.8|72.6±2.1|96.6±0.1|**91.6±0.3**|49.0±2.1|62.0±2.3|72.7±0.9|3.2|
> |UniGCNII|64.3±2.9|57.0±1.2|68.9±3.4|68.6±2.5|74.1±1.5|97.6±0.1|88.3±1.9|32.0±1.5|**77.6±2.0**|76.1±0.9|3.6|
> |AllDeepSets|62.4±3.7|57.8±1.8|67.1±2.2|65.9±1.3|67.6±2.5|96.7±0.3|77.7±5.8|44.0±2.4|73.1±2.2|73.3±2.5|6.6|
> |AllSetTransformer|63.7±1.6|59.5±2.1|69.7±2.3|69.8±2.2|74.2±1.2|96.0±0.2|89.2±0.5|49.3±1.8|76.1±3.8|75.6±2.7|3.4|
> |ED-HNN|64.2±3.0|57.3±1.6|69.6±3.4|69.3±1.6|70.9±1.9|96.7±0.2|90.1±2.1|50.6±3.3|72.3±6.7|71.3±3.7|4.3|
> |ZEN (**ours**)|**67.9±1.4**|**64.0±1.4**|**71.3±1.6**|**72.0±2.0**|73.7±1.2|**98.2±0.1**|88.0±2.9|**51.3±1.1**|74.1±5.2|**76.7±0.4**|**1.9**|
>
> **Response to Question 2:**
>
> > **Question 2:** Could you add the actual running time in ZEN?
>
> Thank you for the suggestion. The actual running times are reported in the table below, with all values measured in seconds (s). ZEN runs in less than a second across all datasets, achieving as fast as 0.003s on the Senate dataset.
>
> |Method|Cora|Citeseer|Pubmed|Cora_ca|20News|MN40|Congress|Walmart|Senate|House|
> |:-|-|-|-|-|-|-|-|-|-|-|
> |HGNN|2.301|2.800|2.423|2.506|2.499|3.958|4.950|20.277|2.540|2.543|
> |HNHN|2.023|1.996|2.877|2.024|3.676|2.720|3.925|14.435|2.278|2.267|
> |HCHA|3.290|3.485|4.448|2.850|2.894|4.072|6.675|8.558|3.297|4.584|
> |UniGCN|4.553|1.932|6.693|2.511|6.825|4.315|8.497|22.121|2.342|1.920|
> |UniGCNII|4.193|3.812|2.069|4.826|4.025|3.831|7.472|24.452|2.277|2.419|
> |AllDeepSets|15.456|18.331|8.889|16.108|8.420|9.378|25.499|60.619|13.232|11.461|
> |AllSetTransformer|2.597|3.167|8.865|3.462|14.273|4.924|6.114|51.702|3.260|3.435|
> |ED-HNN|4.267|4.458|4.407|6.588|11.072|5.664|39.791|31.137|2.336|2.487|
> |ZEN (**ours**)|**0.266**|**0.736**|**0.807**|**0.280**|**0.237**|**0.197**|**0.093**|**0.672**|**0.003**|**0.007**|

---

> > ### Comment · Reviewer_SfcC · 2025-08-03
> >
> > I really appreciate your additional experiments.
> > Your response clarifies my concern; the advantage of a parameter-free model is a small training time, but few-shot learning takes a small training time even in conventionally trained models, so the benefit seems small.
> > For example, even if 0.266 sec becomes 4.267, it may not have a very large impact practically.
> >
> > My score is already high enough, so I will keep my score.

---

> > > ### Author Response · Authors · 2025-08-04
> > >
> > > Thank you again for your thoughtful review and helpful suggestions. Please let us know if any further clarification would be helpful.

---

### Official Review · Reviewer_W1xk · 2025-07-01

**Clarity:** 3
**Significance:** 2
**Originality:** 3
**Rating:** 4
**Confidence:** 3

**Summary:**

This paper focuses on the few-shot node classification task on hypergraphs, and proposes a model, ZEN, to capture high-order relationships while maintaining strong generalization ability and scalability. Experiments on 11 real-world datasets demonstrate the effectiveness and efficiency of the proposed method.

**Questions:**

Q1. More experiments are needed to evaluate the effectiveness of the proposed method under the $n$-way $k$-shot setting.

Q2. The ablation study (Table 6) shows that ZEN underperforms compared to the No TCS variant on the 20News and MN40 datasets, and compared to the No RAP variant on the Senate dataset. Thus, the limitations of ZEN need to be clarified in the revised manuscript.

**Ethical Concerns:**

["NO or VERY MINOR ethics concerns only"]

**Final Justification:**

The authors have conducted additional experiments in the n-way k-shot paradigm and provided a more thorough ablation study. Thus, I will maintain my original score.

**Limitations:**

yes

**Quality:**

3

**Strengths And Weaknesses:**

S1. This paper presents a comprehensive study on liner hypergraph neural networks.

S2. Two techniques, tractable closed-form approximation and redundancy-aware propagation, are developed to improve the efficiency and the corresponding time complexity analysis is provided.

S3. Experiments on real-world datasets have been conducted and the results show the superiority of the proposed method.

W1. While the paper addresses the problem of few-shot node classification, it does not follow the conventional $n$-way $k$-shot learning paradigm in the evaluation setup. This deviation from standard practices should be discussed or justified.

W2. The ablation study shows that removing certain components of ZEN may lead to improved performance, which requires further discussion.

---

> ### Author Rebuttal · Authors · 2025-07-31
>
> Thanks for your thoughtful feedback. We appreciate your constructive review, which has helped us improve the clarity and quality of the paper.
>
> **Response to Weakness 1:**
> > **Weakness 1:** While the paper addresses the problem of few-shot node classification, it does not follow the conventional n-way k-shot learning paradigm in the evaluation setup. This deviation from standard practices should be discussed or justified.
>
> To address the concerns, we would like to respectfully provide clarification on our task formulation. We define our few-shot node classification task as a transductive, semi-supervised node classification under severe label constraints. In standard n-way k-shot learning, the entire set of classes is typically partitioned into disjoint training and test sets. From each set, n classes are sampled, and k labeled examples per class are selected to construct support sets for episodic training. The model is trained on a variety of such tasks in order to evaluate its capacity for adaptation and generalization to novel, unseen tasks. In contrast, our setting can be regarded as a more challenging variant, where the class set remains fixed and identical to the test classes, and n corresponds to the total number of classes.
>
> To avoid potential confusion with n-way k-shot learning from the meta-learning literature, we are open to revising our terminology in the revised version of our paper to reflect this distinction more clearly; e.g., hard few-shot node classification or semi-supervised node classification with hard label scarcity.
>
> On the other hand, it would be interesting to see how our approach fits into the standard n-way k-shot setting. To the best of our knowledge, this setup has not been explicitly explored for general hypergraphs. Accordingly, we designed a new evaluation protocol by drawing inspiration from the experimental setup of Meta-GNN [1] and adopting hyperparameter configurations aligned with those used in ED-HNN [2]. The results are reported in the table below.
>
> |Method|Cora 2-way 3-shot|Cora 2-way 1-shot|Citeseer 2-way 3-shot|Citeseer 2-way 1-shot|
> |:-|-|-|-|-|
> |HGNN|64.4±0.1|55.9±0.1|60.1±0.2|54.9±0.2|
> |UniGCNII|68.0±0.1|58.9±0.2|65.2±0.1|56.8±1.6|
> |AllDeepSets|52.3±0.2|48.7±0.2|51.1±0.2|50.2±0.2|
> |ZEN (**ours**)|**73.3±0.1**|**62.6±0.1**|**71.5±0.1**|**62.3±0.1**|
>
> The strong performance of ZEN in this setting highlights its potential as an effective approach for standard few-shot tasks on hypergraphs. Interestingly, ZEN yields promising results even when applied solely to test episodes, suggesting a potentially valuable and underexplored direction. We appreciate the opportunity to highlight this aspect and will consider it an important avenue for future research.
>
> **Response to Weakness 2:**
> > **Weakness 2:** The ablation study shows that removing certain components of ZEN may lead to improved performance, which requires further discussion.
>
> It is true that removing RAP or TCS leads to performance improvements on some datasets (Congress and Senate for RAP, 20News and MN40 for TCS). To analyze this behavior, we examine the ratio of hyperedges to nodes (i.e., density), which we believe an important factor to understand the characteristic of a dataset. The results are reported in the table below.
>
> |Dataset|Cora|Citeseer|Pubmed|Cora-CA|20News|MN40|Congress|Walmart|Senate|House|
> |:-|-|-|-|-|-|-|-|-|-|-|
> |Density|0.5835|0.3258|0.4041|0.3959|0.0062|1.0000|48.3946|0.7868|1.1160|0.2636|
>
> We observe that RAP improves performance on datasets with low density, while it slightly degrades performance on Congress and Senate, which have densities greater than 1. This aligns with the intuition: as the number of neighbors increases, each node's contribution diminishes due to degree normalization, reducing the relative influence of self-information. In such cases, removing self-information via RAP may have limited or even adverse effects.
>
> As for TCS, it assumes *good* node embeddings. In the ablation study, the impact of removing TCS is more significant on 20News than on MN40. This may due to the extremely low density of 20News, where ZEN, which relies solely on propagation to refine embeddings, may have struggled to produce sufficiently informative representations.
>
> On the other hand, the result on MN40 may have been influenced by its large number of classes — 40, the highest among all datasets. The relative Frobenius norm error of TCS is given by:
>
> $$
> \frac{\left\|\frac{1}{\lambda_1^2}M_1+\frac{1}{\lambda_2^2}M_2+\frac{1}{\lambda_3^2}M_3 - \frac{1}{\epsilon}\left(\frac{1}{\lambda_1}M_1+\frac{1}{\lambda_2}M_2+\frac{1}{\lambda_3}M_3\right)\right\|_F}{\left\|\frac{1}{\lambda_1^2}M_1+\frac{1}{\lambda_2^2}M_2+\frac{1}{\lambda_3^2}M_3\right\|_F}
> $$
>
> As shown in the table below, this approximation error slightly increases with the number of classes:
>
> |$ε$|$c=4$|$c=40$|
> |-|-|-|
> |0.1|1.1064%|1.1753%|
> |0.01|0.1006%|0.1010%|
> |0.001|0.0100%|0.0100%|
>
> Although the error remains small overall, the increase suggests that the approximation may become less accurate as the number of classes grows, which could partially explain the results observed on MN40.
>
> **References**
>
> [1] Fan Zhou, Chengtai Cao, Kunpeng Zhang, Goce Trajcevski, Ting Zhong, and Ji Geng. Meta-gnn: On few-shot node classification in graph meta-learning. CIKM, 2019. URL https://doi.org/10.1145/3357384.3358106.
>
> [2] Peihao Wang, Shenghao Yang, Yunyu Liu, Zhangyang Wang, and Pan Li. Equivariant hypergraph diffusion neural operators, 2023. URL https://arxiv.org/abs/2207.06680.

---

> > ### Comment · Reviewer_W1xk · 2025-08-07
> >
> > Thank you for your response. My concerns have been addressed, so I will not change my score.

---

> > > ### Author Response · Authors · 2025-08-08
> > >
> > > We greatly appreciate your comments. Thank you for the time and effort you dedicated to reviewing our paper.

---

### Official Review · Reviewer_xaQe · 2025-07-02

**Clarity:** 4
**Significance:** 3
**Originality:** 4
**Rating:** 4
**Confidence:** 4

**Summary:**

This paper introduces ZEN (Zero-Parameter Hypergraph Neural Network), a linear and training-free model for few-shot node classification on hypergraphs. ZEN is built upon a unified linearized formulation of existing HNNs and introduces two key innovations: (1) a tractable closed-form solution (TCS) for computing the weight matrix without iterative optimization, and (2) a redundancy-aware propagation (RAP) mechanism to eliminate self-information in multi-hop neighborhoods. The method shows consistent performance improvements over eight strong baselines across 11 real-world datasets. The model is interpretable and the paper includes a detailed ablation and case study.

**Questions:**

Did the authors consider involving more baselines for evaluation?

**Ethical Concerns:**

["NO or VERY MINOR ethics concerns only"]

**Final Justification:**

I appreciate the additional experiments and clarifications, especially the extended baseline comparisons. I will keep my original score.

**Limitations:**

See weaknesses.

**Quality:**

3

**Strengths And Weaknesses:**

Strengths:
1. The formulation of a fully parameter-free HNN using closed-form solutions and a redundancy-aware propagation is both original and impactful, especially for low-resource, few-shot learning scenarios.
2. The theoretical contributions, including the derivation of the closed-form approximation (Theorem 1) and its assumptions, are carefully motivated and reasonably justified.
3. The theoretical contributions, including the derivation of the closed-form approximation (Theorem 1) and its assumptions, are carefully motivated and reasonably justified.

Weaknesses:
1. As acknowledged by the authors, ZEN is specifically designed for node classification. Extensions to other hypergraph tasks (e.g., hyperedge prediction, clustering) are left as future work. This means the scope of applicability of this work is limited. The authors may consider analyzing the effectiveness of the work in other tasks.
2. The paper discusses the success of linear GNNs such as SGC and APPNP, but does not directly compare ZEN to their hypergraph counterparts or generalizations. A direct comparison with a naïve linearized hypergraph model (beyond HGNN) would provide a stronger baseline.
3. ZEN is limited to 2-hop propagation. While this is justified by computational cost and redundancy, it may restrict representational power on deeper hypergraph structures. A discussion on possible approximations or sampling strategies for deeper propagation would be a valuable addition.
4. Lack of baselines. The authors focus on the task of few-shot node classification. However, they only conduct experiments with hypergraph neural networks (HNN) models. There are multiple few-shot node classification works that can act as baselines.

---

> ### Author Rebuttal · Authors · 2025-07-31
>
> Thank you for your valuable comments. Your feedback is greatly appreciated and helps us improve the clarity and quality of our work. We will do our best to address them in the planned revision.
>
> **Response to Weakness 1:**
> > **Weakness 1:** As acknowledged by the authors, ZEN is specifically designed for node classification. Extensions to other hypergraph tasks (e.g., hyperedge prediction, clustering) are left as future work. This means the scope of applicability of this work is limited. The authors may consider analyzing the effectiveness of the work in other tasks.
>
> We fully agree on the importance of assessing our method on other tasks. Some tasks, such as hyperedge prediction, can be reformulated as node classification, allowing ZEN to be applied without modification—for example, by treating hyperedges as nodes using $X_e=D_e^{-1}H^\top X, H_e=H^\top$. For tasks not easily reducible to this setting, ZEN may require further adaptation. We plan to explore such extensions in future work.
>
> **Response to Weakness 2:**
> > **Weakness 2:** The paper discusses the success of linear GNNs such as SGC and APPNP, but does not directly compare ZEN to their hypergraph counterparts or generalizations. A direct comparison with a naïve linearized hypergraph model (beyond HGNN) would provide a stronger baseline.
>
> Thank you for your detailed feedback. We have additionally included three representative linear GNNs (SGC [1], APPNP [2], and SSGC [3]) and three linearized hypergraph models based on UniGCNII [4], AllDeepSets [5], and ED-HNN [6]. For GNNs, we applied clique expansion to convert the hypergraph into a pairwise graph. As shown in the table below, ZEN demonstrates consistently strong performance, outperforming all linear GNN and linearized hypergraph baselines on the majority of datasets.
>
> |Method|Cora|Citeseer|Pubmed|Cora\_ca|20News|MN40|Congress|Walmart|Senate|House|
> |:-|-|-|-|-|-|-|-|-|-|-|
> |SGC|44.1±9.8|40.4±5.3|57.2±5.5|50.8±5.9|58.1±5.1|91.6±0.7|72.5±3.1|24.1±3.2|51.3±2.7|52.3±1.4|
> |APPNP|45.8±9.5|39.6±4.7|55.4±7.1|50.0±6.7|59.9±5.8|92.8±0.3|69.1±6.8|24.7±3.3|71.2±5.4|70.0±8.4|
> |SSGC|42.6±10.9|40.7±4.7|57.1±4.9|52.6±6.7|59.9±5.2|93.8±0.2|81.1±2.4|26.6±3.4|**73.0±0.7**|71.7±3.2|
> |Lin HGNN|42.7±8.4|34.3±9.4|51.7±6.6|49.6±6.9|68.0±6.0|94.5±0.5|83.7±5.0|29.0±6.4|55.8±5.1|57.8±6.5|
> |Lin UniGCNII|39.4±7.4|40.0±4.0|56.9±4.9|49.8±6.1|59.5±5.2|96.6±0.4|73.4±3.1|17.9±1.7|71.2±10.8|68.2±10.6|
> |Lin AllDeepSets|40.5±8.2|40.8±4.5|56.1±6.0|51.0±5.8|66.6±5.0|96.6±0.3|**90.4±1.5**|26.4±3.0|61.0±4.7|65.1±13.1|
> |Lin ED-HNN|42.0±5.6|39.6±4.2|55.8±5.4|47.4±5.8|61.5±6.0|97.3±0.4|78.4±2.7|18.8±1.6|71.1±9.9|70.2±9.3|
> |ZEN (**ours**)|**51.9±10.1**|**49.1±4.8**|**62.6±3.9**|**60.0±6.2**|**68.6±4.8**|**97.6±0.3**|87.0±4.8|**43.9±3.1**|70.4±10.0|**73.2±6.3**|
>
> **Response to Weakness 3:**
> > **Weakness 3:** ZEN is limited to 2-hop propagation. While this is justified by computational cost and redundancy, it may restrict representational power on deeper hypergraph structures. A discussion on possible approximations or sampling strategies for deeper propagation would be a valuable addition.
>
> Thank you for your suggestion. We agree that some datasets may require models with higher-hop propagations for better expressivity. Without approximation, the exact computation of RSI requires high computational cost.
>
> One possible approximation is to estimate the probability that a random walker returns to the starting node after $l$ steps, where $l$ is the number of hops that we aim to model. A simple pseudocode for this strategy is as follows:
> - Start at node $i$.
> - At each step:
>   1. Uniformly sample a hyperedge incident to the current node.
>   2. Uniformly sample a node connected to that hyperedge.
> - Repeat this process for $l$ steps.
> - Record whether the walk ends at the original node $i$.
> The return probability may serve as an approximation of node-wise self-information at a given hop.
>
> Another possible approximation is to use Hutchinson's Estimator, which provides an unbiased stochastic estimate of the diagonal entries. The equation is given by:
> $$
> \operatorname{diag}(A) \approx \frac{1}{m} \sum_{k=1}^m z^{(k)} \odot \left( A z^{(k)} \right)
> $$
> where:
> - $m$ is the number of random probe vectors,
> - each $z^{(k)} \in \mathbb{R}^n$ is a random vector with entries independently sampled from the Rademacher distribution (i.e., each entry is $+1$ or $-1$ with equal probability),
> - $\odot$ denotes the element-wise (Hadamard) product.
>
> The expectation satisfies
> $$
> \operatorname{diag}(A) = \mathbb{E} \left[ z \odot (A z) \right].
> $$
>
> Hutchinson's Estimator does not require explicit storage of the entire matrix $A$; instead, it relies solely on matrix-vector multiplications with randomized vectors $z$. This characteristic renders Hutchinson's Estimator particularly suitable and computationally efficient for hypergraph structures.
>
> We consider these approaches to be promising directions for extending ZEN to scenarios where deeper propagation may provide additional benefits.
>
> **Response to Weakness 4:**
> > **Weakness 4:** Lack of baselines. The authors focus on the task of few-shot node classification. However, they only conduct experiments with hypergraph neural networks (HNN) models. There are multiple few-shot node classification works that can act as baselines.
>
> We appreciate the suggestion to strengthen our evaluation by including additional baselines. To the best of our knowledge, there has been no hypergraph model explicitly addressing few-shot node classification as a direct keyword. In response, we have included two semi-supervised hypergraph models—LEGCN [7] and HyperND [8]—as additional baselines. LEGCN utilizes a line expansion approach to adapt hypergraphs to conventional GNN architectures, while HyperND introduces a diffusion-based mechanism for improved label propagation in hypergraphs. As shown in the table below, ZEN consistently demonstrates strong performance, outperforming all additional baselines across the majority of datasets.
>
> |Method|Cora|Citeseer|Pubmed|Cora\_ca|20News|MN40|Congress|Walmart|Senate|House|
> |:-|-|-|-|-|-|-|-|-|-|-|
> |LEGCN|37.6±5.2|37.3±3.8|58.1±4.0|37.6±5.2|49.4±4.3|93.3±0.7|72.1±3.4|O.O.M|71.3±8.5|72.9±7.0|
> |HyperND|39.1±6.2|35.3±4.8|56.5±4.5|41.7±4.6|54.7±3.8|91.4±0.5|73.6±4.9|13.6±1.7|**73.9±7.5**|72.8±7.1|
> |ZEN (**ours**)|**51.9±10.1**|**49.1±4.8**|**62.6±3.9**|**60.0±6.2**|**68.6±4.8**|**97.6±0.3**|**87.0±4.8**|**43.9±3.1**|70.4±10.0|**73.2±6.3**|
>
> **References**
>
> [1] Felix Wu, Amauri Souza, Tianyi Zhang, Christopher Fifty, Tao Yu, and Kilian Weinberger. Simplifying graph convolutional networks. In PMLR, 2019. URL https://proceedings.mlr.press/v97/wu19e.html.
>
> [2] Johannes Gasteiger, Aleksandar Bojchevski, and Stephan Günnemann. Predict then propagate: Graph neural networks meet personalized pagerank, 2022. URL https://arxiv.org/abs/1810.05997.
>
> [3] Hao Zhu and Piotr Koniusz. Simple spectral graph convolution. In International conference on learning representations, 2021.
>
> [4] Jing Huang and Jie Yang. Unignn: a unified framework for graph and hypergraph neural networks, 2021. URL https://arxiv.org/abs/2105.00956.
>
> [5] Eli Chien, Chao Pan, Jianhao Peng, and Olgica Milenkovic. You are allset: A multiset function framework for hypergraph neural networks, 2022. URL https://arxiv.org/abs/2106.13264.
>
> [6] Peihao Wang, Shenghao Yang, Yunyu Liu, Zhangyang Wang, and Pan Li. Equivariant hypergraph diffusion neural operators, 2023. URL https://arxiv.org/abs/2207.06680.
>
> [7] Yang, Chaoqi and Wang, Ruijie and Yao, Shuochao and Abdelzaher, Tarek. Semi-supervised hypergraph node classification on hypergraph line expansion. CIKM, 2022. URL https://arxiv.org/abs/2005.04843.
>
> [8] Konstantin Prokopchik, Austin R Benson, and Francesco Tudisco. Nonlinear feature diffusion on hypergraphs. PMLR, 2022. URL https://arxiv.org/pdf/2103.14867.

---

> > ### Comment · Reviewer_xaQe · 2025-08-08
> >
> > Thank you for the detailed and thoughtful rebuttal. I appreciate the additional experiments and clarifications, especially the extended baseline comparisons. I will keep my original score.

---

> > > ### Author Response · Authors · 2025-08-08
> > >
> > > Thank you for your valuable feedback. We sincerely appreciate the time and effort you put into reviewing our work.

---

### Author Response · Authors · 2025-08-05

We sincerely appreciate the reviewers’ insightful comments, which have greatly helped improve our submission. We have made every effort to address the concerns raised and would be happy to provide further clarification should any questions remain.

---

### Note · Authors · 2025-08-12

Dear Reviewers and ACs,

We sincerely thank you for your time, effort, and constructive feedback, which have greatly improved the quality of our work. We’re encouraged that reviewers recognized our study’s contributions, particularly noting:
- The original and impactful (xaQe) formulation and comprehensive (W1xk) investigation of a linear, fully parameter-free HNN.
- Solid, rigorous (UPmr), carefully motivated, and well-justified (xaQe) theoretical contributions.
- Surprising (SfcC) performance across ten datasets, demonstrating the superiority of the proposed method (W1xk).
- A well-written paper (UPmr) with sufficient validation and strong theoretical support (SfcC).

We have carefully addressed all concerns through additional experiments and analyses:
- Generality (xaQe, W1xk, SfcC): Evaluated 10-shot, 20-shot, and n-way k-shot settings with consistent outperformance of ZEN. Demonstrated that tasks such as hyperedge prediction can be reformulated as node classification.
- Error Bound (UPmr): Derived the relative Frobenius norm error, confirming it remains sufficiently low as ε decreases.
- Baselines (xaQe, SfcC): Added eight competitive baselines—three linear GNNs, three linearized HNNs, and two standard HNNs—with ZEN outperforming them across settings.
- Higher-hop Approximation (xaQe): Explored random-walk and Hutchinson’s estimator as approximation strategies.
- Ablation Studies (W1xk): Analyzed the effects of graph density, error norm, and class number.
- Real-time Efficiency (SfcC, UPmr): Reported actual running times as low as 0.003 seconds on the Senate dataset.

We are deeply grateful for the opportunity to refine our work through this process. The constructive input from the reviewers and ACs has significantly strengthened our study and will guide our future research.

Best regards,

Authors

---

### Decision · Program_Chairs · 2025-09-17

**Decision:**

Accept (poster)

**Comment:**

This paper introduces ZEN, a parameter-free hypergraph neural network for few-shot node classification. ZEN is built on a unified linearized formulation of HNNs. Two key novelties include a tractable closed-form solution for weight computation and a redundancy-aware propagation mechanism to avoid self-information leakage. Extensive experiments across 11 real-world datasets show consistent improvements over strong baselines.

The main strengths lie in its originality and simplicity: the parameter-free formulation is theoretically grounded and empirically outperforms previous hypergraph neural networks. The paper is also clearly written, with thorough ablations and case studies.

The weaknesses are relatively minor. The method is currently limited to node classification; broader applicability to other hypergraph tasks remains unexplored. The evaluation setup only partially follows the conventional few-shot learning paradigm, which could be better justified. ZEN is restricted to 2-hop propagation, which may limit representation capacity, and some comparisons (e.g., with more few-shot methods or deeper hypergraph variants) are missing. Finally, further clarification on time complexity, error bounds, and reporting conventions would improve the presentation.

Overall, despite these weaknesses, the paper makes a solid and meaningful contribution to hypergraph learning. Its theoretical insights, empirical effectiveness, and efficiency justify acceptance. I recommend acceptance.